# A library of base editors for the precise ablation of all protein-coding genes in the mouse mitochondrial genome

Pedro Silva-Pinheiro ©, Christian D. Mutti ©, Lindsey Van Haute, Christopher A. Powell, Pavel A. Nash, Keira Turner © & Michal Minczuk © ✉

The development of curative treatments for mitochondrial diseases, which are often caused by mutations in mitochondrial DNA (mtDNA) that impair energy metabolism and other aspects of cellular homoeostasis, is hindered by an incomplete understanding of the underlying biology and a scarcity of cellular and animal models. Here we report the design and application of a library of double-stranded-DNA deaminase-derived cytosine base editors optimized for the precise ablation of every mtDNA protein-coding gene in the mouse mitochondrial genome. We used the library, which we named MitoKO, to produce near-homoplasmic knockout cells in vitro and to generate a mouse knockout with high heteroplasmy levels and no off-target edits. MitoKO should facilitate systematic and comprehensive investigations of mtDNA-related pathways and their impact on organismal homoeostasis, and aid the generation of clinically meaningful in vivo models of mtDNA dysfunction.

Mammalian mitochondria contain several copies of their own genome (mtDNA) which encodes 13 essential subunits of the oxidative phosphorylation (OXPHOS) system. Pathogenic variants in the mitochondrial genome (both 'heteroplasmic' (mix of normal and mutant genomes) or 'homoplasmic' (100% mutant genomes)) can result in mitochondrial diseases, which are a major group of inherited conditions affecting ~1 in 8,000 humans[1]. These disorders are characterized by unexplained tissue selectivity and are currently incurable. Accumulation of mtDNA mutations has also been observed in healthy humans, in particular during the ageing process, and has been associated with common multifactorial diseases, metabolic disease, heart failure, cancer and neurodegeneration[2]. There is a pressing need to develop new approaches to model mtDNA dysfunction in vitro and in vivo, which will be indispensable for experimental therapy development and, in the longer term, to treat diseases in which mtDNA dysfunction is a primary or confounding factor[3].

Historically, the mitochondrial biology and medicine fields were unable to manipulate or modify the mitochondrial genome in mammalian mitochondria within cells, which has severely hindered research on mtDNA metabolism and the development of in vivo models for pre-clinical therapies for mtDNA diseases[4]. Only a few mouse models for mtDNA disease have been developed and characterized thus far[5]. For many years, the approaches towards manipulation of mtDNA in mammals have been mainly limited to changing the ratio of the existing mtDNA variants (heteroplasmy) by mitochondrially targeted restriction enzymes and programmable nucleases, both in vitro[6–10] and in vivo[11–14]. A method to silence mitochondrial gene expression in a systematic way was developed, utilizing chemically synthesized precursor-morpholino hybrids[15]. However, this strategy is only applicable in purified mitochondria, invaliding its use for studies in cellular and animal models. Recently, a new approach has been developed: DddA-derived cytosine base editor (DdCBE), which catalyses site-specific C:G to T:A conversions in mtDNA. DdCBE is based on an adapted toxin DddA_{tox} derived from *Burkholderia cenocepacia* (separated non-toxic halves fused to TALE proteins), which is targeted to the mitochondrial matrix to catalyse deamination of cytidines within double-stranded DNA at a sequence determined by TALE design in vitro and in vivo[16]. The initial DdCBEs deaminate cytidines in

MRC Mitochondrial Biology Unit, University of Cambridge, Cambridge, UK. ✉e-mail: michal.minczuk@mrc-mbu.cam.ac.uk

the TC:GA sequence context leading to a TC:GA > TT:AA[16]. Very recently, engineered zinc finger-based mitochondrial deaminases have also been developed[17] and the strict TC sequence-context constraint of DddA$_{tox}$ was expanded to offer a broadened HC (H = A, C or T) sequence compatibility[18].

Recent reports on the use of DdCBE in plants showed relatively high efficiency of editing of plant mitochondrial and plastid genomes, but a substantial burden of off-target edits was also observed[19–21]. A proof-of-concept of successful installation of mtDNA edits in animals in vivo has recently been provided, by delivering DdCBE-coding nucleic acids into embryos in mice[22,23], rats[24] and zebrafish[23], and by adeno-associated virus–mediated delivery into post-mitotic tissues in mice[25]. However, off-target mutations in the mitochondrial genomes have been observed and none of these animals have shown high mtDNA editing levels (heteroplasmy, let alone homoplasmy), raising some concerns about the specificity and efficacy of DdCBEs for in vivo use.

Here we report a library of improved mitochondrial base editors (MitoKO) for systematic and comprehensive investigations of mtDNA-related pathways. MitoKO provides an easy access solution for researchers who require the ablation of mtDNA-encoded protein-coding genes in the mouse. The MitoKO library can be applied to study fundamental processes occurring in mitochondria and their impact on organismal homoeostasis, and to generate novel, clinically meaningful in vivo models of mtDNA dysfunction for drug discovery and pre-clinical investigations.

## Results

### Design of MitoKO DdCBE constructs

We set out to generate the MitoKO—a library of highly specific DdCBEs to knock out (KO) every protein-coding gene of mouse mtDNA through the introduction of premature stop codons. For each open reading frame (ORF), we designed eight DdCBE pairs containing TALE domains binding the mtDNA light (L) or heavy (H) strands (DdCBE-L1 and DdCBE-L2 or DdCBE-H1 and DdCBE-H2, respectively) and different combinations of the 1333 DddA$_{tox}$ split (1333 N or 1333 C) targeting a 14–20-bp-long sequence in the mitochondrial ORFs (Fig. 1a and Extended Data Fig. 1a,b). For all mtDNA-encoded mouse ORFs, except *MT-Nd4l*, we intended to change Trp codons TGA into TAA STOP codons by deaminating the C on the opposite (non-coding) strand (5′ TCA > 5′ TTA; edited C underlined) (Fig. 1b). In the case of *MT-Nd4l*, we changed a coding sequence for Val90 and Gln91 (GTC CAA) into Val and STOP (GTT-TAA; edited Cs underlined) by deaminating two consecutive Cs on the coding strand (Fig. 1c). These designs led to a collection of truncating mutations as early as at the 6th amino acid-coding codon *(MT-CoI)* and not later than the 146th amino acid-coding codon *(MT-CoIII)* (Fig. 2a,b).

### Screening MitoKO DdCBE pairs

We screened the MitoKO designs using transient high-level expression of DdCBEs. To this end, DdCBEs were cloned into vectors that co-express fluorescent marker proteins (mCherry or GFP) enabling fluorescence-activated cell sorting (FACS) of transiently transfected cells (Extended Data Fig. 1c). In the initial screen, we intended to optimize the 1333 DddA$_{tox}$ split orientation. We tested MitoKO DdCBEs with the N-terminal part of the 1333 DddA$_{tox}$ split (1333 N) linked either with L-strand-binding TALEs (DdCBE-L1 or DdCBE-L2) or H-strand-binding TALEs (DdCBE-H1 or DdCBE-H2) in combination with the C-terminal part of the 1333 DddA$_{tox}$ split (1333 C) linked with pairing/matching TALE constructs (Fig. 3a and Extended Data Fig. 1a,b). We transfected these constructs into NIH/3T3 mouse cells, which were subjected to FACS at 24 h post-transfection to enrich the population of cells expressing the designated MitoKO DdCBEs. After FACS, we seeded the transfectants for continued culture and collected them at 7 d post-transfection for mtDNA heteroplasmy analyses (Extended Data Fig. 1c). The 1333 DddA$_{tox}$ split orientation screen revealed that linking 1333 C with L-strand binding TALEs led

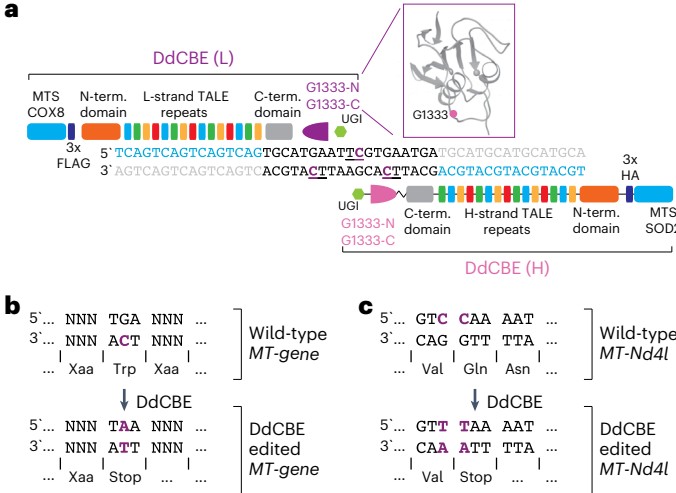

**Fig. 1 | MitoKO DdCBE library design strategy. a**, The architecture of DdCBE monomers used to generate the mitoKO library targeting each protein-coding gene of the mouse mtDNA. The mtDNA specificity is provided by programmable TALE domains. In each experiment, the DddA$_{tox}$ G1333 split (purple) was tested in both orientations to achieve editing of the desired 'TC' sites. The MTS were from human superoxide dismutase 2 (SOD2) or cytochrome C oxidase subunit 8 (COX8). UGI, uracil glycosylase inhibitor. L-strand or (L), light mtDNA strand. H-strand or (H), heavy mtDNA strand. **b**, The strategy employed to ablate the 12 out of 13 mtDNA protein-encoded genes (*Nd1, Nd2, Nd3, Nd4, Nd5, Nd6, Cytb, CoI, CoII, CoIII, Atp6* and *Atp8*) by introducing a premature stop codon with base editing. In the vertebrate mitochondrial genetic code, the TGA codon encodes tryptophan (Trp). Transition of the cytosine (C) in the opposite strand to a thymine (T) (in purple) using base editing leads to a premature TAA stop codon. **c**, The strategy employed to knock out the mtDNA protein-encoded gene *Nd4l*, by introducing a premature stop codon with base editing. Transition of the cytosine (C) in a CAA codon encoding glutamine (Gln) to a thymine (T) (in purple) using base editing leads to a premature TAA stop codon, thereby silencing the *Nd4l* gene. In this site, base editing can potentially edit the adjacent C of a GTC codon encoding valine (Val). However, the resulting GTT codon also encodes valine leading to a silent mutation.

to the expected C editing of the TGA Trp codon on the opposite non-coding H-strand for *Nd1, Nd2, Nd3, Nd4, Nd5, Cytb, CoI, CoII, CoIII, Atp6* and *Atp8* (Fig. 3b and Extended Data Fig. 1a,b). For *Nd6*, which is encoded by the H-strand, the opposite was observed—linking 1333 C with H-strand binding TALEs led to the expected C editing of the TGA Trp codon on the opposite non-coding L-strand (Fig. 3b). Finally, for *Nd4l* (expected edits: GTC CAA > GTT TAA; edited C underlined), linking 1333 C with H-strand binding TALEs also led to higher on-target editing levels (Fig. 3b). Having established the favourable DddA$_{tox}$ 1333 split orientation, we set out to optimize the position of TALE domain binding for each DdCBE pair combined with scoring off-target effects (Fig. 3c). The score used to assess off-targets introduced a penalty score for mtDNA off-targets with heteroplasmy greater than 5% (Supplementary Dataset 1). This analysis led to the selection of lead pairs for each of the mtDNA-encoded ORFs (Fig. 3d, arrow heads). The on-target activity of the lead MitoKO DdCBE pairs ranged between ~40% and ~70% (Fig. 3e), which was higher than the originally reported DdCBEs, which ranged between ~5 and ~50%[16]. This higher editing efficacy could be explained by differences in DdCBE construct selection strategies (antibiotic vs FACS) or in cell type used (human HEK293 vs mouse NIH/3T3, which can substantially differ, for example, in terms of mitochondrial import or mtDNA base excision repair efficiencies). Taken together, we have generated a library of 13 DdCBEs capable of introducing high levels of premature stop codons into the ORFs of mouse mtDNA, therefore, knocking out each of the mtDNA-encoded protein-coding genes.

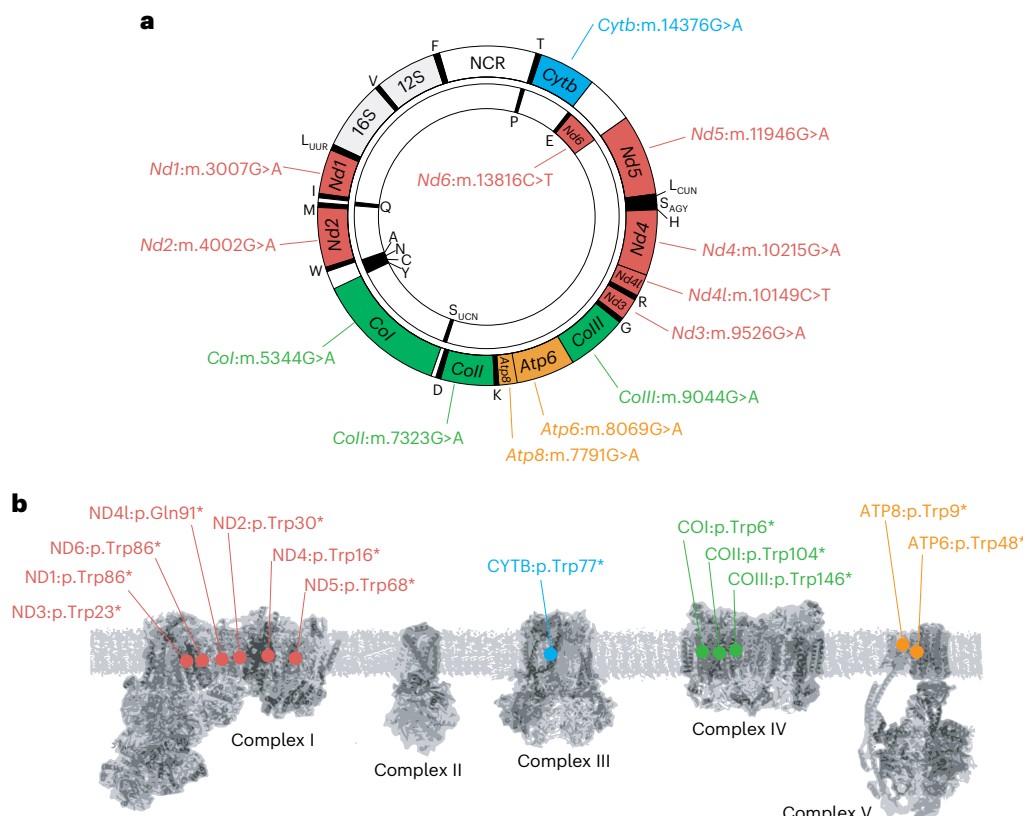

**Fig. 2 | Target sites of the MitoKO DdCBE library. a**, Genetic map of mouse mtDNA indicating the position of DdCBE-introduced STOP mutations. **b**, Schematic representation of the OXPHOS system indicating the positions of protein truncation.

## Generation of near-homoplasmic mtDNA knockouts by iterative application of MitoKO constructs

Next, we set out to generate high levels of truncating mutations in all mtDNA-encoded ORFs by repeated MitoKO treatments. To this end, we performed several sequential rounds of transfection and recovery, consisting of MitoKO construct delivery and selection of transfectants by FACS at 24 h, followed by a 14 d recovery period, at which point heteroplasmy was measured and cells were re-transfected (Fig. 4a,b). Beginning with wild-type (WT) NIH/3T3 cells and using four iterative cycles of transfection and recovery, this approach was capable of producing effectively homoplasmic cells harbouring premature STOP codons in each of the mouse mtDNA-encoded protein-coding genes (Extended Data Fig. 2). To substantiate the genetic data showing complete knockout of the mtDNA-encoded ORFs, we assessed mitochondrial translation of each of the 13 mtDNA-encoded polypeptides by incorporating radioactively labelled methionine upon inhibition of cytoplasmic protein synthesis. This analysis revealed that the mitochondrial de novo protein synthesis rate was markedly decreased for the genes harbouring a MitoKO-introduced STOP mutation, supporting precise installation of the gene-disrupting edits (Fig. 4c and Extended Data Fig. 3a). As specific bands for Nd4l, Nd5, Nd6, CoIII and Atp8 are not easily identifiable in the de novo protein synthesis assay, we examined whether their genetic ablation results in perturbation of steady-state levels of the corresponding OXPHOS complexes using blue-native gel electrophoresis (BNGE) (the reduction of mtDNA-encoded core subunits of OXPHOS complexes generally leads to their aberrant assembly and/or instability[26]). This analysis showed substantially reduced levels of complex I in *Nd4l*, *Nd5* and *Nd6* knockouts, complex III in *CoIII* knockout and complex V in *Atp8*-ablated cells (Extended Data Fig. 3b). Consistent with this, basal oxygen consumption rates were significantly reduced in all 13 mtDNA knockout cell lines, except for the *Nd4* KO line, for which

oxygen consumption rates were lower although without statistical significance (Extended Data Fig. 4a). Also, the growth of these 13 mtDNA knockout cell lines on galactose-containing medium, which forces the cells to rely on OXPHOS to produce ATP, was severely compromised (Extended Data Fig. 4b,c). Interestingly, continuous culture of MitoKO cell lines on galactose-containing medium for more than 7 d led to a partial loss of the damaging nonsense mutations (Extended Data Fig. 4d). Taken together, these results demonstrate that sequential short-term MitoKO DdCBE treatments could achieve near-complete knockouts of mtDNA proteins, thus enabling systematic interrogation of the mouse OXPHOS system using reverse genetics.

## Limiting off-target mutagenesis with improved MitoKO architectures

We then focused on assessing and optimizing the precision of the MitoKO, with the aim of bringing the off-target mutagenesis to background levels found in wild-type cells. To compare mtDNA-wide off-targeting of the MitoKO DdCBEs with the previously published mitochondrial cytosine base editors[16], we re-analysed mtDNA from the WT NIH/3T3 cells and cells transfected with MitoKO DdCBEs (14 d post-transfection). Wild-type cells were used as a control to distinguish MitoKO DdCBE-induced C:G-to-T:A single-nucleotide variants (SNVs) from natural background heteroplasmy, which was at 0.02%. The average frequencies of mtDNA-wide off-target C:G-to-T:A editing by MitoKO DdCBEs were between ~3.5 to ~14.5-fold higher (0.065–0.29%) than background heteroplasmy frequency for the control (Extended Data Fig. 5a,b). While for many MitoKO constructs the mtDNA-wide off-target frequencies were well within the range of 'precise' DdCBEs (0.04–0.15%)[16], for some of them the off-targets were higher than the values observed for the originally reported DdCBEs editing mtDNA in human HEK293T cells[16]. Furthermore, there was a positive correlation

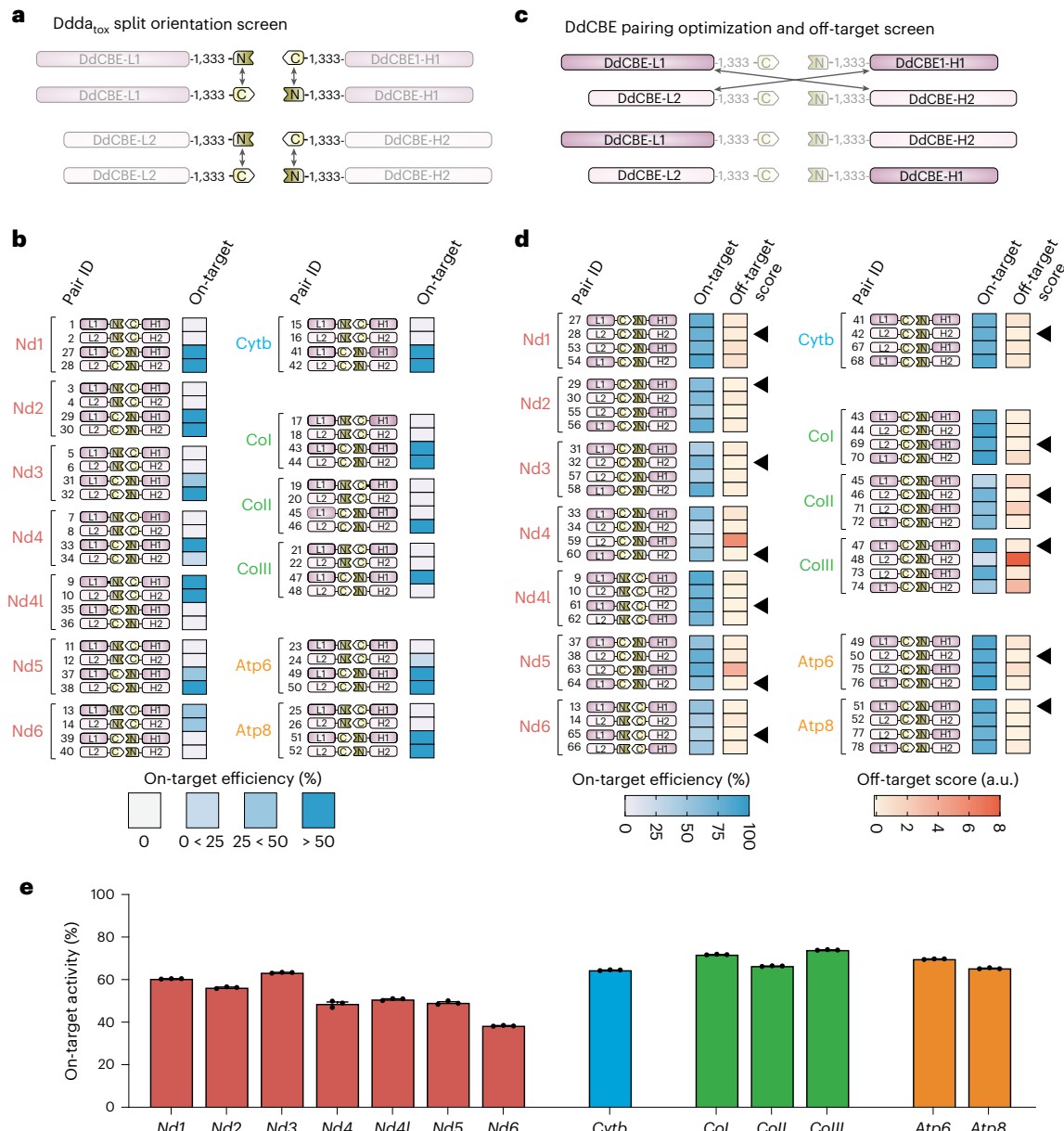

**Fig. 3 | Selection screening of DdCBEs for MitoKO library generation. a,** Schematic of the first screening for the optimization of DddA$_{tox}$ G1333 split orientation. In this screen, DdCBEs were generated by pairing TALEs L1 with H1 and L2 with H2 (Extended Data Fig. 1), testing the N-terminal fragment of DddA$_{tox}$ G1333 linked to L1 or L2 and the C-terminal part linked to H1 or H2, and the reciprocal orientation, with the C terminal of DddA$_{tox}$ G1333 linked to L1 or L2 and the N terminal linked to H1 or H2. **b,** In vitro on-target editing efficiency in cells after transient expression with the indicated MitoKO DdCBE pairs (Pair ID). The schematics represent the DdCBE pair combinations and which DddA$_{tox}$ G1333 split orientation was used. On-target efficiency was analysed by Sanger sequencing. Note the impact of split orientation on the on-target efficiency. **c,** Schematic of the second screening for the optimization of DdCBE pairings and

off-target analysis. In this screen, DdCBEs with the most efficient DddA$_{tox}$ G1333 split orientation (from screen 1; **b,c**) were tested with additional pairings: TALEs L1 with H2 and L2 with H1. **d,** In vitro on-target editing efficiency and off-target scores in NIH/3T3 cells after transient expression with the indicated DdCBE pairs from the MitoKO library (Pair ID). The schematics represent the DdCBE pair combinations and which DddA$_{tox}$ G1333 split orientation was used. On-target efficiency and off-targets were analysed by next-generation sequencing (NGS). The final MitoKO DdCBEs (indicated with black arrows) were selected on the basis of high on-target efficiency and low off-target score (less mtDNA-wide off-targets). Source data are provided as a Source Data file. **e,** On-target editing by the lead MitoKO DdCBE pairs from 14 d after transfection measured by NGS. Bars and error bars represent mean ± s.e.m. ($n$ = 3 technical replicates).

between on-target and off-target mtDNA editing, with increased levels of STOP mutants being accompanied by higher levels of C:G-to-T:A SNVs (Extended Data Fig. 5c). Given the latter, we hypothesized that reducing DdCBE expression levels would lead to an improved on-target/off-target ratio. Therefore, we sought to engineer new DdCBE architectures to exercise greater control over protein expression levels. To this end, we considered three approaches: (1) The inclusion of an engineered hammerhead ribozyme (HHR) into the 3′ untranslated region (UTR) of

MitoKO DdCBE messenger RNA to constitutively cleave coding mRNA, leading to a poly(A)-free 3′ end that is susceptible to degradation, hence greatly reducing protein expression[27] (Fig. 5a). (2) Linking the MitoKO DdCBE monomer coding sequences with the T2A element, therefore greatly reducing the mitochondrial concentration of the downstream monomer[28] (Fig. 5b). (3) Combining the HHR and T2A approaches (Fig. 5). Our previous studies showed that the inclusion of HHR 3′K19 element[27] into the 3′ UTR of engineered mitochondrially-targeted zinc

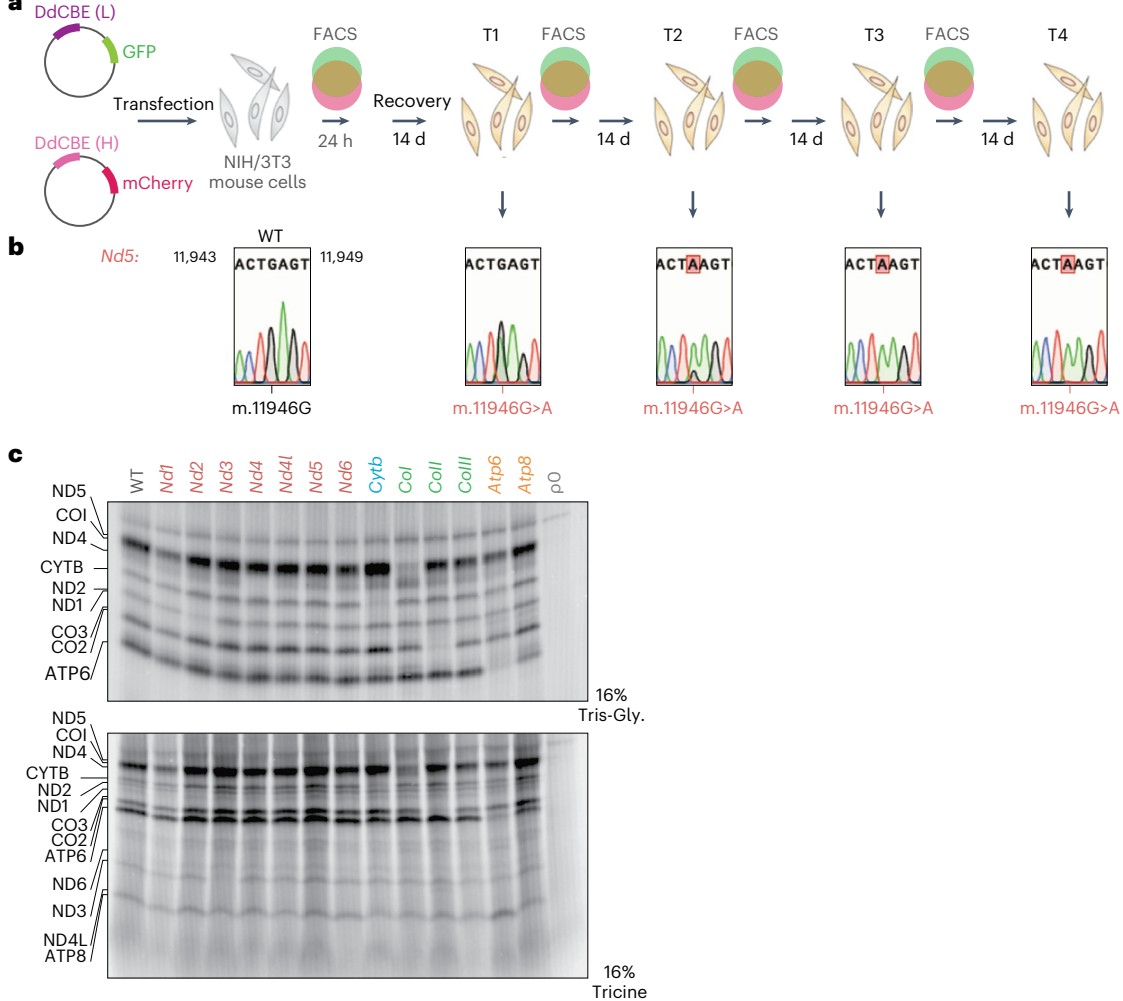

**Fig. 4 | Near-homoplasmic mtDNA editing using the MitoKO library.**
**a**, Schematic of the general workflow for evaluation of the four iterative transfection and recovery cycles (T1–T4) of MitoKO pairings (for further details, see Extended Data Fig. 1c). **b**, Sanger sequencing of the *Nd5* editing site upon iterative cycles of expression and recovery (for the remaining mtDNA genes, see Extended Data Fig. 2). **c**, Mitochondrial de novo protein synthesis in the NIH/3T3

cells that underwent four rounds of MitotKO DdCBE treatment, assessed by ³⁵S-methionine metabolic labelling. WT and mtDNA-less Rho-0 cells (p0) were used as controls. The mtDNA-encoded products were resolved in two gel systems (16% Tris-Gly and 16% Tricine). For densitometric quantification, see Extended Data Fig. 3a.

finger nucleases led to the reduction of their expression (Fig. 5a) and a consequent decrease in nuclease off-targeting[29]. We included 3′K19 HHR between the DdCBE ORF stop codon and the bovine growth hormone (BGH) poly(A) site of four MitoKO constructs, targeting subunits of each OXPHOS complex that contains mtDNA-encoded components (complex I, *Nd3*; complex III, *Cytb*; complex IV, *CoII*; complex V, *Atp6*) (Fig. 5a,c). We delivered these constructs with improved DdCBE architecture as separate plasmids into WT NIH/3T3 cells, selected the transformants by FACS and assessed their on- and off-target performance (Fig. 5c, left, 2-plasmid or 2-plasmid-HHR). The HHR-equipped MitoKO DdCBEs showed greatly reduced off-targeting, as assessed by analysing mtDNA-wide C:G-to-T:A SNVs (Fig. 5c, Extended Data Fig. 6 and Supplementary Dataset 2), with the mtDNA off-targeting for *Nd3* and *Cytb* MitoKO constructs being comparable to that of the WT control, and mtDNA off-targeting for *CoII* and *Atp6* being only ~1.5-fold higher than that of the control (Fig. 5c, 2-plasmid-HHR). Next, we used the T2A sequence to tandemly link the ORFs coding for the MitoKO DdCBE monomers ablating *Nd3, Cytb, CoII* and *Atp6*, delivered these constructs on single plasmids into WT NIH/3T3 cells, selected the transformants by FACS (mCherry selection) and assessed their mitochondrial on- and off-target performance (Fig. 5c, left, Tandem). This tandem architecture

also yielded a substantial reduction in off-targeting as scored by assessing mtDNA-wide C:G-to-T:A SNVs. The off-targets for *Nd3, Cytb* and *Atp6* were virtually indistinguishable from those of the control, while the *CoII* construct off-targeting was ~1.5-fold higher than for WT NIH/3T3 cells (Fig. 5c, Extended Data Fig. 6 and Supplementary Dataset 2, Tandem). Finally, WT NIH/3T3 cells were transfected with the T2A-linked and HHR-equipped *Nd3, Cytb, CoII* and *Atp6* MitoKO constructs, followed by FACS and mtDNA-wide C:G-to-T:A off-target SNV assessment (Fig. 5c, left, Tandem-HHR). The combination of the T2A and HHR architectures led to further reduction in mtDNA off-targeting for all analysed constructs, with the average C:G-to-T:A off-target SNVs being at the level of WT NIH/3T3 cells (Fig. 5c, Extended Data Fig. 6 and Supplementary Dataset 2, Tandem-HHR).

According to recent data, high levels of DdCBE expression led to off-target editing in the nuclear genome (nDNA) in human cultured cells[30] and mouse embryos[31]. Therefore, we intended to assess whether MitoKO also leads to nDNA off-targets and, if that were the case, to observe whether the improved T2A-based DdCBE architecture leads to a reduction in nDNA off-targeting. To this end, we performed whole-genome sequencing in wild-type NIH/3T3 cells and cells transfected with the *Atp6* MitoKO construct expressed either from two

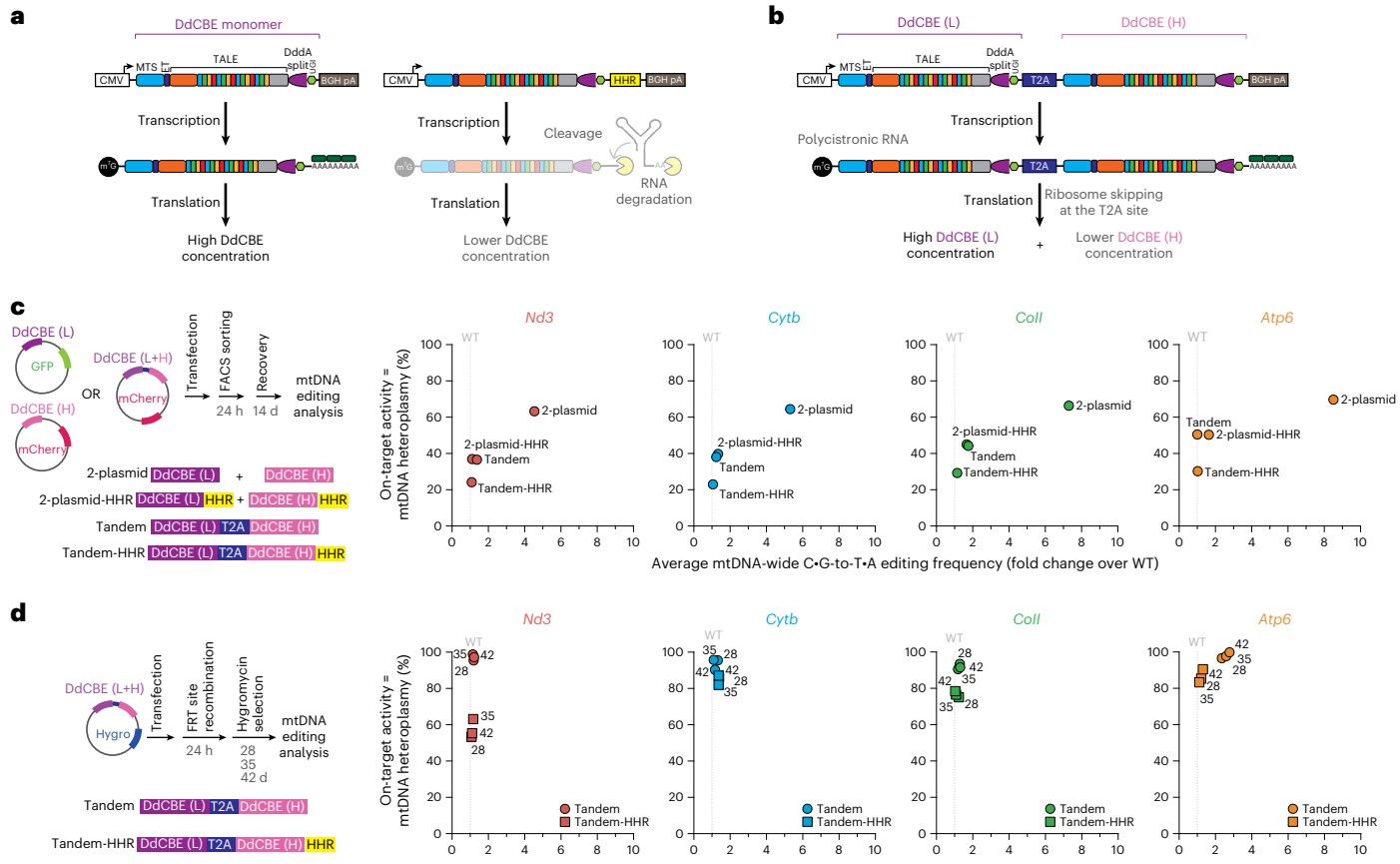

**Fig. 5 | Optimization of MitoKO constructs by limiting off-target mutagenesis. a**, Schematic of MitoKO transgene (DdCBE monomer) without (left) or with the HHR incorporated (right). After transcription, the mRNA encoding DdCBE is constitutively degraded following HHR cleavage, resulting in substantially lower quantities of translated protein. ET, epitope tag; BGH pA, bovine growth hormone polyadenylation signal. Other symbols/abbreviations as in (Fig. 1a.) or defined in text. **b**, Schematic of MitoKO transgenes (DdCBE (L) and DdCBE (H)) linked by the T2A ribosome skipping sequence. Expression of the DdCBE in the tandem T2A-linked arrangements leads to lower concentrations of the downstream monomer in the mitochondrial matrix. **c**, Left: schematic of the general workflow for the on/off-target optimization experiments, involving transient transfection of NIH/3T3 cells with plasmids co-expressing DdCBE monomers and fluorescent marker proteins, FACS-based selection of cells expressing both monomers, recovery and phenotypic evaluation of DdCBE-treated cells (top). Schematic representation of the DdCBE arrangements used

(bottom). Right: on-target (Y axis) and off-target (X axis) performance of the Nd3, Cytb, CoII and Atp6 MitoKO constructs transiently delivered into NIH/3T3 cells as separate monomers (2-plasmid), separate monomers with the HHR incorporated in mRNA (2-plasmid-HHR), bi-cistronic construct with the tandemly arrayed DdCBE monomers being linked by the T2A element (Tandem) and the tandem T2A-linked monomers harbouring HHR (Tandem-HHR). Dots represent the mean (n = 3 biological replicates). Source data are provided as a Source Data file. **d**, Left: schematic of the workflow of the on/off-target optimization experiments for stably expressed tandem constructs without (Tandem) or with HHR (Tandem-HHR), involving transfection, recombination into a docking nuclear DNA site and hygromycin selection (top). Schematic representation of the DdCBE arrangements used (bottom). Right: on-target (Y axis) and off-target (X axis) performance of the Nd3, Cytb, CoII and Atp6 Tandem and Tandem-HHR MitoKO constructs stably expressed in NIH/3T3 cells. Dots represent the mean (n = 3 biological replicates).

separate plasmids (Fig. 5c, left, 2-plasmid) or from a single construct (Fig. 5c, left, Tandem). First, we selected 109 C:G-to-T:A SNVs from five chromosomes (chr15 to chr19) that differed the most between WT and 2-plasmid Atp6 MitoKO samples and from those we analysed 14 and 7 SNVs in the TC and TCC sequence context, respectively. None of these SNVs showed statistically significant differences between the WT and Atp6 MitoKO 2-plasmid or WT and Atp6 MitoKO Tandem conditions (Extended Data Fig. 7). The lack of detectable off-targets compared with published studies in human cells[30] or mouse embryos[31] could be attributable to differences in cell type used and/or DdCBE expression levels.

Taken together, fine-tuning expression levels of MitoKO DdCBEs with an upgraded architecture through the inclusion of an HHR in the 3′ UTR and/or tandemly linking the DdCBE monomers using the T2A element has led to great improvement in base editing precision by reducing mtDNA off-targeting effects of our MitoKO constructs to background levels.

## Generation of near-homoplasmic mtDNA knockouts by long-term application of MitoKO constructs

The improvement in the MitoKO library precision led to 2–3-fold reduction in on-target performance (Fig. 5c, Extended Data Fig. 6 and Supplementary Dataset 2). We hypothesized that long-term expression of the tandem and Tandem-HHR MitoKO architectures (Fig. 5c) would lead to an increased on-target base editing while maintaining the same precision levels. To this end, we used a docking site in the genome of the NIH/3T3 mouse cell line for FRT recombinase-assisted insertion of the T2A-linked and/or HHR-containing Nd3, Cytb, CoII and Atp6 MitoKO constructs, followed by hygromycin selection of stable transfectants (Fig. 5d, left). Then we analysed on- and off-target mtDNA base editing at 28, 35 and 48 d post-transfection. For all T2A-linked architectures tested, we observed near-homoplasmic introduction of premature stop codons (Fig. 5d, Tandem), while off-targeting remained low and comparable to the WT cell background (Fig. 5d and Supplementary Dataset 2, Tandem), except for the Atp6 MitoKO construct (Fig. 5d, orange

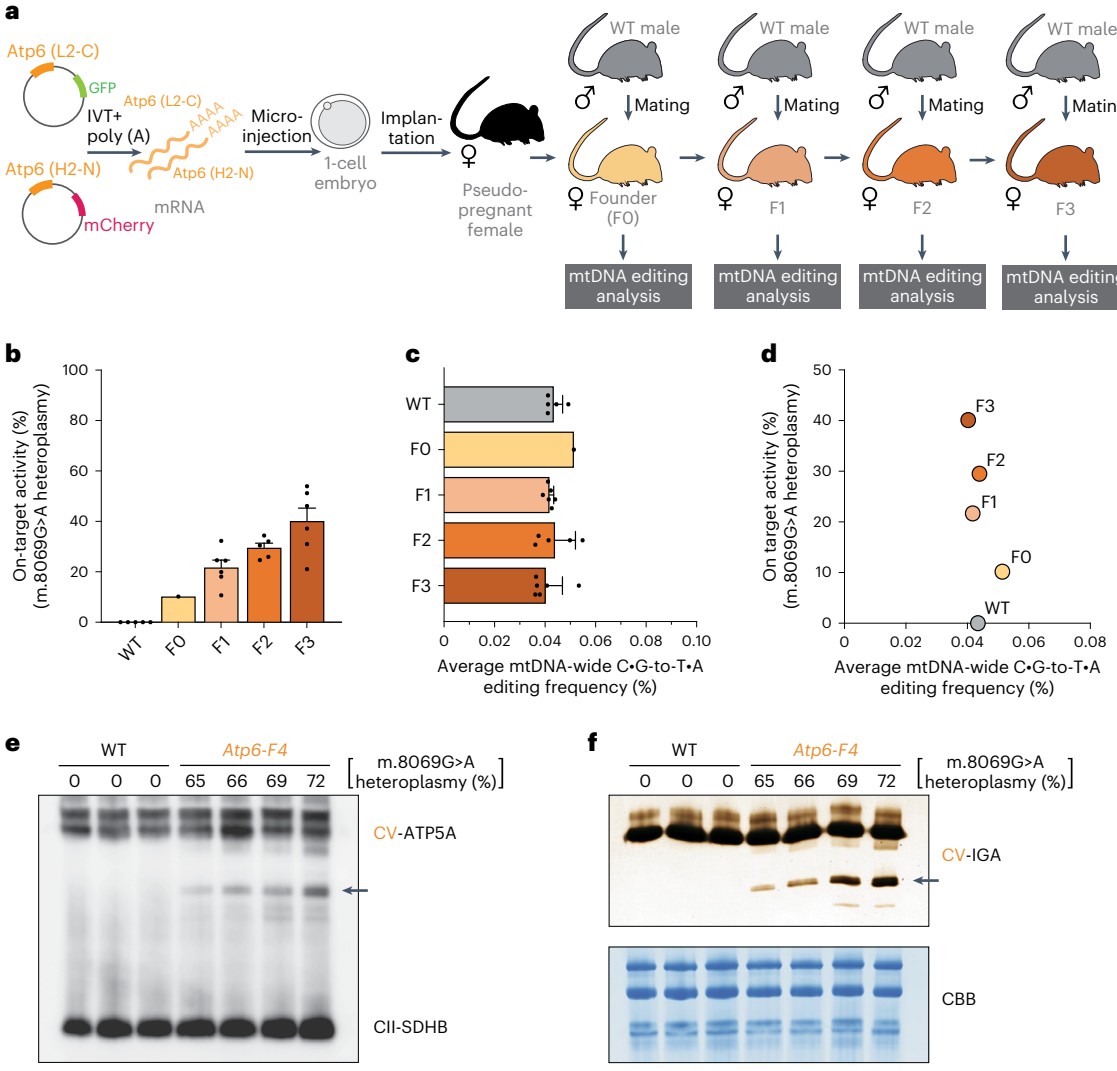

**Fig. 6 | In vivo mouse mtDNA editing with the MitoKO library. a**, Schematic representation of the use of Atp6 MitoKO constructs (L2-C and H2-N) for the generation of F0 founder animals carrying a heteroplasmic m.8069 G > A mutation and subsequent selective breeding scheme. IVT+poly(A), in vitro transcription and polyadenylation. **b**, The m.8069 G > A heteroplasmy (Y axis) in the F1–F3 generations of the selectively bred founder F0 female. Bars and error bars represent the mean ± s.e.m. **c**, Off-target editing in the F0–F3 m.8069 G > A animals and WT controls. Bars and error bars represent the mean ± s.e.m. **d**, On-target (Y axis) vs off-target (X axis) editing in the F0–F3 m.8069 G > A animals and WT controls. **e**, Immunoblotting of mitochondria isolated from skeletal muscle of F4 generation Atp6 m.8069 G > A heteroplasmic mice and WT controls upon a CNGE with a CV-specific antibody (ATP5A). Arrow indicates CV subcomplexes. An antibody specific for complex II (SDHB) was used as loading control. **f**, CNGE followed by in-gel ATP hydrolysis activity of CV in mitochondria isolated from skeletal muscle of F4 generation Atp6 m.8069 G > A heteroplasmic mice and WT controls. Arrow indicates the activity of CV subassemblies. Coomassie brilliant blue (CBB) was used as loading control. For **e** and **f**, uncropped scans are provided as a Source Data file.

circles). Similarly, long-term expression of most of the T2A-linked and HHR-containing MitoKO architectures (Fig. 5d, Tandem-HHR) led to high heteroplasmies of the premature stop codons (>80%), except for *Nd3*, with minimal off-target detection (Fig. 5d and Supplementary Dataset 2, Tandem-HHR). Taken together, we generated mouse cell lines with high heteroplasmy levels of protein-coding gene knockouts and very low off-target editing by long-term expression of either T2A (*Nd3*, *Cytb, CoII*) or T2A/HHR (*Atp6*) improved MitoKO library constructs.

## In vivo mouse mtDNA editing with the MitoKO library

Next, we intended to test the utility of the MitoKO library in modifying the mitochondrial genome in vivo. Several truncating mutations in *Mt-Atp6* have been associated with mitochondrial disease with a heterogenous clinical spectrum and various tissue-specific presentations[32–36]. Therefore, we decided to model these mitochondrial ATP synthase (complex V or CV) disorders in mice. To do this, we injected

1-cell embryos with in vitro transcribed (IVT), polyadenylated mRNA coding for the MitoKO DdCBE introducing m.8069 G > A nonsense mutation in *Atp6* (Fig. 6a). Next, we implanted DdCBE-injected embryos into pseudopregnant, surrogate mothers and obtained F0 offspring carrying approximately 3% (female), 10% (female) and 3% (male) of m.8069 G > A heteroplasmy. To verify germline transmission of the resulting mitochondrial mutation, the female F0 mouse carrying 10% of m.8069 G > A was crossed with a wild-type C57BL6/J male to obtain F1 pups (Fig. 6a and Supplementary Dataset 3). The first litter of the F1 generation pups carried m.8069 G > A heteroplasmy ranging between 9% to 23% (Fig. 6b and Supplementary Dataset 3). Next, we selectively bred the highest heteroplasmy females from F1 and F2 to obtain animals harbouring more than 50% m.8069 G > A heteroplasmy in F3 (Fig. 6b and Supplementary Dataset 3). We also carried out off-targeting analysis of the generations between F0 and F3 by analysing mtDNA-wide C:G-to-T:A SNVs and comparing to WT animals. While the F0 female

showed a slightly elevated level of C:G-to-T:A SNVs, these got purified in the subsequent generations (Fig. 6c). Importantly, the average m.8069 G > A heteroplasmy values, gradually rising through generations due to selective breeding, were not accompanied by any detectable increase in C:G-to-T:A SNVs, consistent with no linked off-targeting being carried through the maternal line (Fig. 6d). Having established that F1–F3 generations have virtually no off-targets, we further bred the mice to produce F4 and obtained higher m.8069 G > A heteroplasmy levels (Supplementary Dataset 3). We then analysed CV in skeletal muscle of F4 mice by immunoblotting. Similar to patients carrying truncating *Mt-Atp6* mutations[36], we observed CV subcomplexes in the heteroplasmic F4 mice, as detected using an antibody to ATP5A, the early assembled ATP synthase F1 subunit α (Fig. 6e). An ATP hydrolysis in-gel activity assay confirmed that the subassemblies are consistent with the F1 extramembranous catalytic core of CV (Fig. 6f). It is worth noting that the levels of CV subcomplexes positively correlated with m.8069 G > A heteroplasmy (Fig. 6e,f). Taken together, we show that the MitoKO library constructs can be used for in vivo modelling of mitochondrial dysfunction and we report a mouse model with ATP synthase dysfunction due to an mtDNA mutation.

## Discussion

We developed and optimized MitoKO to enable systematic reverse engineering of the mitochondrial genome in mouse cells. The MitoKO is a library of highly specific DdCBEs to knock out every protein-coding gene of mouse mtDNA through the introduction of premature stop codons. The MitoKO constructs are extensively optimized for on-target activity and show virtually no off-target action, forming a collection of reagents for: (1) in vitro use in cultured mouse cells via convenient single-plasmid delivery; (2) in vivo applications for embryo microinjections, embryonic stem cell transfection or hydrodynamic delivery; and (3) a collection of knockout mouse cell lines in the NIH/3T3 background with isogenic distribution of cells with variable mtDNA mutant levels from the same parental source without clonal selection, which is normally required for the generation of cytoplasmic hybrid (cybrid) lines and can lead to a substantial nuclear genome variability[37]. The MitoKO repository will be constantly updated, with the reagents being available to the scientific community. In the next stage, we plan to implement the potential of somatic tissue-specific mutagenesis with adeno-associated virus and Rosa26/TIGRE locus transgenesis for spatiotemporal expression control.

While testing on the MitoKO library in vitro, we observed a positive correlation between on-target and off-target editing, with increased levels of on-target editing being accompanied by higher levels of mtDNA-wide C:G-to-T:A off-targets (Extended Data Fig. 5c). Fine-tuning the expression levels of the MitoKO constructs through architecture improvement led to low levels of off-targets, thus enabling high precision of the MitoKO constructs. Limiting the concentration of DdCBEs in the mitochondrial matrix was pursued with two architectures: (1) the incorporation of HHR that reduces DdCBE mRNA stability, hence reducing levels of translated protein and (2) linking of DdCBE monomers with a T2A peptide. While use of T2A peptides necessitates the addition of a C-terminal peptide tail to the upstream protein and an N-terminal prolyl addition to the downstream protein, these were shown not to appreciably affect the on-target deaminase activity. We also speculate that the N-terminal proline functions as a partial mitochondrial targeting sequence (MTS) 'mask', diminishing the rate of downstream mitochondrial protein import.

While progress has been made in evolving the protein-only cytosine base editor DdCBE to overcome the strong TC sequence context dependency of DddA[18], these developments are probably more useful for precise modelling and correction of disease-associated mtDNA point mutation in human cells at non-TC target sites. However, for the development of the MitoKO library, the strict TC requirement was useful as we could select targeting windows with a single (or a limited

number of) cytosine in the correct sequence context, hence improving targeting specificity. While with optimization of MitoKO we achieved off-target editing at the level of controls, further work could aim at optimizing the interface contact between split DddA$_{tox}$ fragments to limit the formation of active deaminase with two adjacent DNA-binding domains (ZF or TALE) being bound to mtDNA, similar to the previous work done with obligatory heterodimeric programmable nucleases[38].

Two recent reports showed that non-optimized DdCBEs cause numerous off-target editing for C-to-T/G-to-A conversions in the nuclear genome in human cells[30] or mouse embryos[31], which in the case of mouse embryos were twice as frequent compared with off-targets introduced by low-fidelity CRISPR/Cas9-based cytosine base editor[31]. While we did not observe any substantial nDNA off-targets for MitoKO *Atp6* constructs (Extended Data Fig. 7), these previous results highlight the necessity of extensive DdCBE optimization and development of methods allowing for detection of DdCBE-induced nuclear off-target editing in large cell populations. However, we note that in the context of in vivo use of the MitoKO library, this is less of an issue for two reasons: (1) Mitochondria containing precise edits can be transferred to embryonic stem cells for mouse model generation as done routinely in the past[39] and (2) such potential nuclear off-targets can be purified through back-crossing with wild-type males during the selective breeding procedure reported in this work, since mtDNA is maternally transmitted. Also, for any in vitro work with cultured cells, potential nuclear off-targets can be avoided by producing cybrids, whereby mitochondria harbouring MitoKO DdCBE-edited mtDNA are transferred to mtDNA-less recipient cells that have not undergone DdCBE treatment, using standard protocols[40,41].

The MitoKO library also overcomes the limitation of the recent study that used precursor-morpholino chimera imported into mitochondria for silencing of mitochondrial gene expression. While this strategy can be useful to harness mechanistic questions regarding mitochondrial gene expression, it can only be applied in vitro in mitochondria isolated from cells[15].

In summary, here we provide a precise and universal toolset for systematic functional interrogation of the OXPHOS system in vitro and in vivo, enabling the development of pre-clinical models of mitochondrial dysfunction. We envisage that the MitoKO library reported here will be widely used for basic and biomedical research and to underpin the notion of the role of mitochondria in aging, cancer and neurodegenerative diseases.

## Methods

### Ethics statement

All animal experiments were carried out in accordance with the UK Animals (Scientific Procedures) Act 1986 (Procedure Project Licence: P6C20975A) and EU Directive 2010/63/EU, and authorized by the University of Cambridge Animal Welfare and Ethical Review Body.

### Plasmid construction

The DdCBE architectures used were as previously reported[16]. TALE arrays were designed using the Repeat Variable Diresidues (RVDs) containing NI, NG, NN and HD amino acids, recognizing A, T, G and C, respectively. To construct the plasmids used in the cell screens, all DdCBEs ORFs were synthesized as gene blocks (GeneArt, Thermo Fisher). DdCBEs targeting the L-strand of the mtDNA were cloned in a pTracer cytomegalovirus promoter (CMV)/Bsd (pTracer) backbone which co-expresses eGFP, while DdCBEs targeting the H-strand of the mtDNA were cloned in a pcDNA3.1(−) mCherry (pcmCherry) backbone which co-expresses mCherry. DNA targeting sequences of all DdCBEs used in this study can be found in Extended Data Fig. 1b. Amino acid sequences of the final selected pairs can be found in Supplementary Sequences 1. To generate the tandem architectures, the selected DdCBE (L) was amplified by PCR to include 5′ NotI and 3′ NheI while also removing the 3′ stop codon, and the selected DdCBE (H) was amplified by PCR

to include a 5′ NheI and 3′ KpnI while also including a 5′ T2A sequence to promote ribosome skipping. The two fragments DdCBE (L) and (H) were ligated using the NheI site and in the same reaction cloned in tandem into a pcmCherry backbone using the 5′ NotI and 3′ KpnI sites. The singleHHR and Tandem-HHR versions were generated by incorporating a 3K19 HHR sequence at the 3′ end of the ORF, as previously described[13]. Vectors used for stable expression were prepared by 'copy-pasting' the Tandem and Tandem-HHR DdCBEs described above into a modified pcDNA5/FRT (Thermo Fisher, V601020) backbone using the 5′ NotI and 3′ KpnI sites. Modifications of the pcDNA5/FRT backbone include the exchange of the CMV promoter by an EF-1α promoter and the incorporation of the DddI$_A$ gene from *B. cenocepacia* under the pre-existent Lac promoter using the BstZ17I site. The DddI$_A$ gene was found to be necessary possibly to mitigate bacterial toxicity caused by cloning both DddA halves (1333 N and 1333 C) in a single vector. Flp recombinase expression for specific genomic integration of the DdCBE expression cassette was accomplished using the Flp recombinase expression plasmid, pOG44 (Thermo Fisher, V600520).

## Cell culture and transfections

NIH/3T3 (CRL-1658, American Type Culture Collection (ATCC)), Flp-In-3T3 (Thermo Fisher, R76107) and 3T3 rho-0#8 (Kerafast, ESA101) cells were cultured at 37 °C under 5% (v/v) CO$_2$ in complete Dulbecco's modified Eagle medium (DMEM) (4.5 g l$^{-1}$ glucose, 2 mM glutamine, 110 mg ml$^{-1}$ sodium pyruvate), supplemented with 10% calf bovine serum with iron, 1% penicillin/streptomycin (all from Gibco) and 50 μg ml$^{-1}$ uridine (Sigma, U3750). Mycoplasma tests in the culture medium were negative. The cell lines were not authenticated in this study. For DdCBE pair screens, NIH/3T3 mouse cells plated in 6-well tissue culture plates at a confluency of 70% were transfected with 3,200 ng of each monomer (L and H), or 6,400 ng of plasmid DNA when using Tandem architectures in a single plasmid (L + H), with 16 μl of FuGENE-HD (Promega) following the manufacturer's guidelines. After 24 h, cells were collected for FACS and sorted for GFP and mCherry double-positive cells (or just mCherry in Tandem experiments) using a BD FACSMelody cell sorter. The collected cells were allowed to recover for another 6 d (or as indicated in the figure legend) and then used for DNA extraction as described below. For generation of stable cell lines expressing Tandem and Tandem-HHR DdCBEs, Flp-In-3T3 mouse cells were transfected with 3,200 ng of the expression vector, together with 3,200 ng of the Flp recombinase expression plasmid (pOG44), using 19 μl FuGENE-HD (Promega) following the manufacturer's guidelines. After 24 h, the cells were exposed to selection media supplemented with 200 μg ml$^{-1}$ hygromycin B (Thermo Fisher, 10687010). Cells resistant to hygromycin B were expanded and DNA was extracted for further analysis at 28, 35 and 42 d after transfection.

## Oxygen consumption measurements

Oxygen consumption rates were measured using an XF24 extracellular flux analyser (Seahorse Biosciences). NIH/3T3 cells were seeded at a density of 25,000 cells per well in 250 μl of culture media in an XF 24-well cell culture microplate (Seahorse Biosciences) and incubated for 6 h at 37 °C in 5% (v/v) CO$_2$. The culture medium was then replaced with 250 μl of bicarbonate-free DMEM and cells were incubated at 37 °C for 30 min before measurement.

## Measurement of cell growth

For cell growth assays, DdCBE-transfected NIH/3T3 and 3T3 rho-0#8 cells were grown in either glucose-containing DMEM (4.5 g l$^{-1}$ glucose, 110 mg l$^{-1}$ sodium pyruvate, 10% FBS, 100 U ml$^{-1}$ penicillin, 100 μg ml$^{-1}$ streptomycin) or galactose-containing DMEM (0.9 g l$^{-1}$ galactose, 110 mg l$^{-1}$ sodium pyruvate, 10% FBS, 100 U ml$^{-1}$ penicillin, 100 μg ml$^{-1}$ streptomycin). Confluency was measured using an Incucyte S3 live-cell imaging system (Essen BioScience). These measurements were taken at ×4 zoom, with 9 images per well every 6 h.

## $^{35}$S-methionine labelling of mitochondrial translation products

Labelling of newly synthesized mitochondrially expressed proteins was performed as previously described[42]. Briefly, DdCBE-treated cells at approximately 80% confluency were incubated in methionine/cysteine-free medium for 10 min. The medium was then replaced with methionine/cysteine-free medium containing 10% dialysed FCS and emetine dihydrochloride (100 μg ml$^{-1}$) to inhibit cytosolic translation. Following a 20 min incubation, 120 μCi ml$^{-1}$ of ($^{35}$S)-methionine was added and the cells were incubated for 60 min. After washing with 1X PBS, cells were lysed, and 30 μg of protein was loaded on either 16% Tris-glycine or 16% Tricine SDS–PAGE gels, as indicated in the figure. Dried gels were visualized with a PhosphorImager system (Amersham Typhoon 5 scanner). Densitometric quantification was performed using ImageJ along the midpoint of each lane and plotted using GraphPad Prism.

## mRNA preparation and microinjection of mouse zygotes

The Atp6 (L2-C) and Atp6 (H2-N) DdCBE mRNAs were synthesized using the in vitro RNA transcription kit mMESSAGE mMACHINE t7 Ultra kit (Ambion), utilizing the pre-existing 5′ end T7 promoter in the pTracer and pcmCherry backbones. The resulting polyadenylated mRNAs were purified with a MEGAclear kit (Ambion). For microinjection, 300 ng ml$^{-1}$ Atp6 (L2-C) and 300 ng ml$^{-1}$ Atp6 (H2-N) mRNAs were diluted in microinjection buffer (5 mM Tris-HCl, pH 7.4, 0.1 mM EDTA, pH 8.0) and injected into the cytoplasm of mouse 1-cell embryos. Superovulation, embryo collection, microinjections and implantation in pseudopregnant females were carried out by the in-house transgenic services at the University of Cambridge, UK, following standardized protocols.

## Animals

Atp6-KO heteroplasmic mice were kept on a C57BL/6J background. The animals were maintained in a temperature- and humidity-controlled animal care facility with a 12 h light/12 h dark cycle and free access to water and food. Genotyping and off-target analyses of new litters were carried out using ear biopsies collected when weaning at approximately 3 weeks of age.

## Genomic DNA isolation and Sanger sequencing

Cells were collected by trypsinization, washed once in PBS and resuspended in lysis buffer (1 mM EDTA, 1% Tween 20, 50 mM Tris (pH 8)) with 200 μg ml$^{-1}$ proteinase K. Lysates were incubated at 56 °C with agitation (300 r.p.m.) for 1 h, and then incubated at 95 °C for 10 min before use in downstream applications. Genomic DNA from mouse ear biopsies was extracted with a Maxwell 16 tissue DNA purification kit in a Maxwell 16 instrument (Promega), according to the manufacturer's instructions.

For Sanger sequencing, a ~300 bp region from each mtDNA protein-coding gene was PCR-amplified with Go*Taq* G2 DNA polymerase (Promega) using specific primers indicated in Supplementary Table 1. The PCR was performed with an initial heating step of 1 min at 95 °C, followed by 35 cycles of amplification (30 s at 95 °C, 30 s at 62 °C, 15 s at 72 °C) and a final step of 5 min at 72 °C. PCR purification and Sanger sequencing were carried out by Source Bioscience (UK) with the corresponding mtDNA gene primer indicated in Supplementary Table 1.

## High-throughput targeted amplicon mtDNA sequencing, processing and mapping

For mtDNA-wide sequence analysis, two overlapping long amplicons (8,331 bp and 8,605 bp) covering the full mtDNA molecule were amplified by long-range PCR with PrimeSTAR GXL DNA polymerase (TAKARA) using the following primers: mmu_ND2_Fw: 5′- TCT CCG TGC TAC CTA AAC ACC -3′; with mmu_ND5_Rv: 5′- GGC TGA GGT GAG GAT AAG CA -3′; and mmu_ND2_Rv: 5′- GTA CGA TGG CCA GGA GGA TA -3′; with mmu_ND5_Fw: 5′- CTT CCC ACT GTA CAC CAC CA -3′. The PCR was performed with an initial heating step of 1 min at 94 °C, followed by

16 cycles of amplification (30 s at 98 °C, 30 s at 60 °C, 9 min at 72 °C) and a final step of 5 min at 72 °C. Tagmentation and the indexing PCR were performed using the Nextera XT index kit (Illumina, FC-131-1096) according to the manufacturer's instructions. Briefly, the indexing PCR was performed with an initial heating step of 30 s at 98 °C, followed by 12 cycles of amplification (30 s at 98 °C, 30 s at 55 °C, 30 s at 72 °C) and a final step of 5 min at 72 °C. Libraries were subjected to high-throughput sequencing using the Illumina MiSeq or NovaSeq platform (PE250) and demultiplexed using the Illumina MiSeq or NovaSeq manufacturer's software. For processing and mapping of high-throughput data related to mtDNA-wide analysis, a quality trimming and 3′ end adaptor clipping of sequenced reads were performed simultaneously, using Trim Galore! (--paired)[43]. For mtDNA-wide sequence analysis, reads were aligned to ChrM of the mouse reference genome (GRCm38) with Bowtie2 (--very-sensitive; --no-mixed; --no-discordant)[44]. Count tables were generated with samtools mpileup (-q 30)[45] and varscan[46].

### mtDNA off-target scoring
To comparatively assess the different DdCBE pair combinations, an off-target score was generated. The score penalizes for higher C:G-to-T:A off-target frequencies and compensates for higher base editing on target. The calculation was as follows: the sum of all C:G-to-T:A off-target frequencies over 5% plus 25 times the sum of all C:G-to-T:A off-target frequencies over 20%, divided by the on-target frequency squared.

### Whole-genome sequencing, processing and mapping
For whole-genome sequencing analysis, the fragmentation/end prep, adaptor ligation and indexing PCR were performed using the NEBNext Ultra II FS DNA library prep kit for Illumina (NEB, E6177) according to the manufacturer's instructions, with 25 ng of genomic DNA as initial input. Briefly, the fragmentation step was performed for 10 min at 37 °C. The indexing PCR was performed with an initial heating step of 30 s at 98 °C, followed by 13 cycles of amplification (10 s at 98 °C, 75 s at 65 °C) with a final step of 5 min at 72 °C. Libraries were subjected to high-throughput sequencing using the Illumina NovaSeq platform (PE150) and demultiplexed using the Illumina NovaSeq manufacturer's software. For processing and mapping of high-throughput data, adapter trimmed reads[43] were aligned to the mouse genome with HiSat2 (--no-spliced-alignment –maxins 700 –no-mixed –no-discordant) (PMID: 31375807). Mapped reads were extracted per chromosome with Samtools[45] and variant calling was performed with samtools mpileup and Varscan[46]. Only genome positions with a read depth of more than 30 in all samples were used in further analysis. All C:G-to-T:A SNVs that had 45% of variants in WT samples were removed, as they most probably represent differences (hetero or homozygous) in the NIH/3T3 genome as compared with the mouse reference genome or positions that are difficult to sequence. C:G-to-T:A SNVs with significant confidence between WT and DdCBE-treated samples were detected on the basis of a two-way analysis of varianve (ANOVA). Further analysis was performed for SNVs identified in the DdCBE preferred sequence context: 5′-TC-3′ and 5′-TCC-3′.

### Blue-native gel electrophoresis (BNGE), clear-native gel electrophoresis (CNGE) and in-gel activity analysis
Samples for BNGE were prepared from digitonized cellular extracts, while samples for CNGE were prepared from isolated skeletal muscle (quadriceps) mitochondria as previously described[47]. For solubilization, the samples were resuspended in 1.5 M aminocaproic acid, 50 mM Bis-Tris/HCl (pH 7) and 1.6 mg dodecyl maltoside per mg of protein, and incubated for 5 min on ice before centrifuging at 20,000 × g at 4 °C. The supernatants were collected to new tubes and NativePAGE 4X sample buffer (Thermo Fisher, BN2003) was added to each sample to a final concentration of 1X. For the BNGE samples, 5% Coomassie G250 was also added.

Samples for both BNGE and CNGE were loaded onto NativePAGE 3–12% Bis-Tris Gels (Thermo Fisher) and electrophoresis was performed using NativePAGE Running Buffer system (Thermo Fisher). The BNGE cathode buffer was supplemented with 20X NativePAGE cathode buffer. The CNGE cathode buffer, the 20X NativePAGE cathode, was substituted by 10% dodecyl maltoside and 10% sodium deoxycholate. The gels were run at a constant 90 V for 30 min, followed by 12 mA for 3 h at 4 °C. After electrophoresis, transfer to a PVDF membrane was performed using a wet system with a constant current of 300 mA at 4 °C. The membranes were blocked in 5% milk in PBS with 0.1% Tween 20 (PBS-T) for 1 h at room temperature and then incubated overnight at 4 °C with antibodies specific to each mitochondrial complex: mouse anti-NDUFB8, 1:1,000 (Abcam, ab110242); mouse anti-SDHB, 1:2,000 (Abcam, ab14714); mouse anti-UQRC2, 1:1,000 (Abcam, ab14745); mouse anti-COX IV, 1:1,000 (Abcam, ab14744) and mouse anti-ATP5A, 1:1,000 (Abcam, ab14748) all diluted in 5% milk. Membranes were then washed three times with PBS-T for 10 min at room temperature and then incubated with an HRP-linked secondary antibody anti-mouse IgG (Promega, W4021) diluted 1:2,500 in 5% milk in PBS-T. The membranes were washed another three times as before and imaged digitally with an Amersham Imager 680 blot and gel imager (GE Healthcare) upon incubation with Amersham ECL western blotting detection reagents (GE Healthcare).

The in-gel ATP hydrolysis activities of complex V and dissociated F1-subcomplex were analysed by incubating the CNGE gels with 35 mM Tris-HCl pH 7.8, 270 mM glycine, 14 mM MgSO4, 0.2% Pb(NO$_3$)$_2$ and 8 mM ATP. The lead phosphate precipitation that is proportional to the enzymatic ATP hydrolysis activity was stopped by 50% v/v methanol after an overnight incubation, and the gels were then transferred to water until imaging.

### Statistical analysis
Graphical visualization of data and all statistical analyses were performed with GraphPad Prism (version 9.1.0). All numerical data are expressed as mean ± s.e.m. Ordinary one-way ANOVA with Dunnett's test and two-way ANOVA with either Sidak's or Tukey's test were used for multiple comparisons as specified in the figure legends.

### Reporting summary
Further information on research design is available in the Nature Portfolio Reporting Summary linked to this article.

## Data availability
The data supporting the findings of this study are available within the paper and its Supplementary Information. The NGS files generated in this study are available from the GEO database via the accession number GSE202643. Source data for the figures are provided with this paper.

## Code availability
MtDNA on and off-target effects from next-generation sequencing data were calculated with Varscan2 (source code available at https://github.com/Jeltje/varscan2; ref. [46]). Proprietary scripts are available from the corresponding author on reasonable request.

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

## Acknowledgements

We acknowledge the members of the Mitochondrial Genetics Group (MRC-MBU, University of Cambridge) for useful discussion during the course of this research. We thank J. Carroll for help in obtaining the data for Fig. 6f. All authors disclose funding support for the research described in this study from Medical Research Council UK (MC_UU_00015/4 and MC_UU_00028/3). P.S-P. discloses support for the research described in this study from The Champ Foundation (G112428). P.A.N. discloses support for the research described in this study from The Lily Foundation (G101554).

## Author contributions

P.S-P. and M.M. planned and designed experiments; C.D.M. performed the FACS and was involved in the NGS experiments; L.V.H. analysed the NGS experiments; C.A.P. performed the $^{35}$S-methionine labelling experiments; P.A.N. performed the genotyping of the Atp6-KO mice; K.T. managed the animal work. P.S-P. performed the remaining experiments. P.S-P. and M.M. drafted the manuscript. M.M. supervised the study. All authors revised the manuscript.

## Competing interests

M.M. is a founder, shareholder and member of the Scientific Advisory Board of Pretzel Therapeutics, Inc. P.S-P., P.A.N., L.V.H., and C.D.M. provide consultancy services for Pretzel Therapeutics. P.S-P., C.D.M and M.M. have filed patent applications on this work. L.V.H. is director of NextGenSeek Ltd. The remaining authors declare no competing interests.

## Additional information

**Extended data** is available for this paper at https://doi.org/10.1038/s41551-022-00968-1.

**Correspondence and requests for materials** should be addressed to Michal Minczuk.

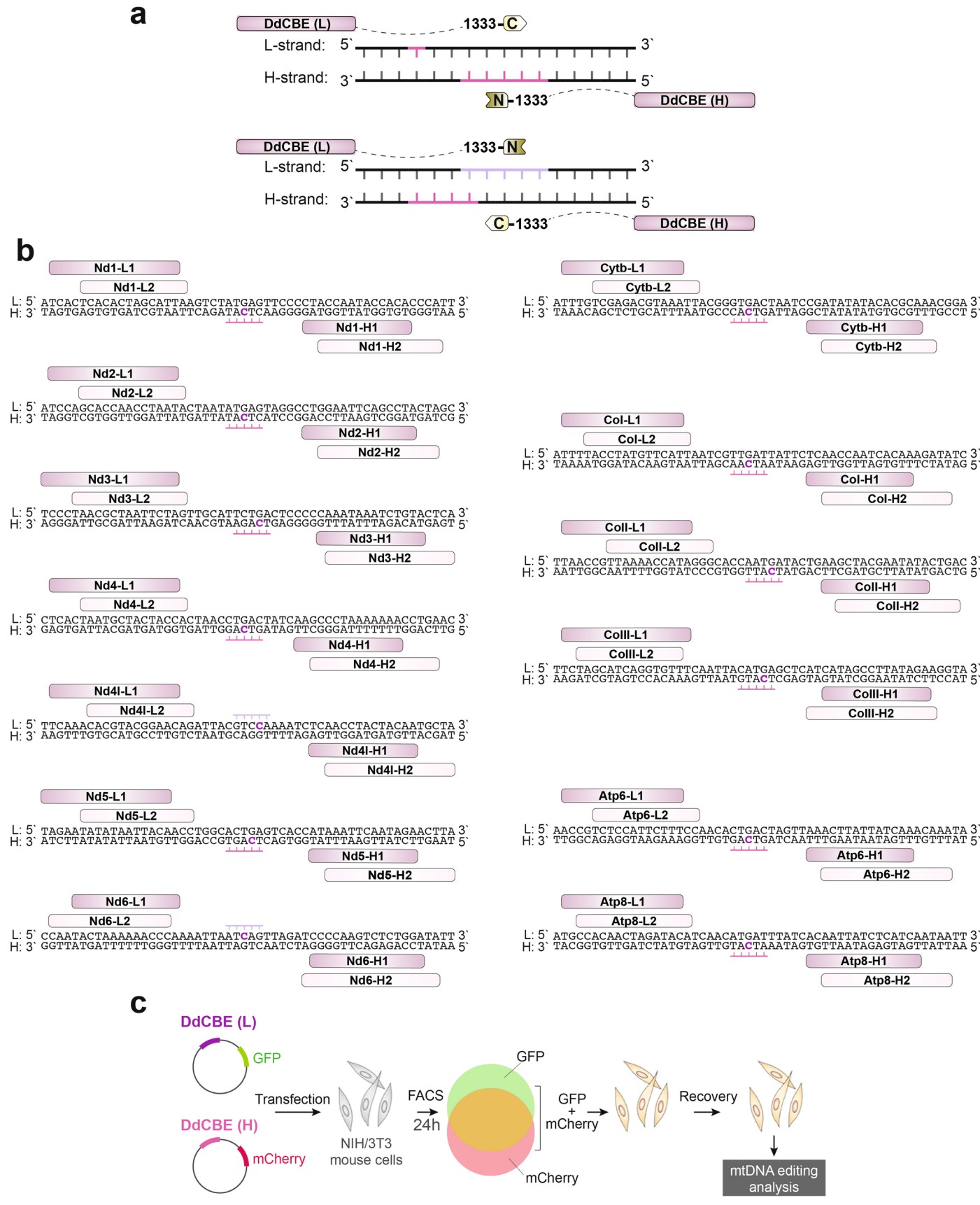

**Extended Data Fig. 1 | See next page for caption.**

**Extended Data Fig. 1 | Design of the MitoKO library. a**. Schematic representation of possible DdCBE orientations of the DddA$_{tox}$ G1333 split and corresponding preference for base editing of cytosines within the DNA targeting window. The N and C-terminal fragments of the DddA$_{tox}$ G1333 split can be linked to the TALE arrays binding either L-strand or H-strand of the mtDNA. The split orientation influences which cytosines are edited. **b**. The TALE designs to target all mouse protein-coding genes in the mouse mitochondrial genome. For each mtDNA protein-encoding gene, two TALEs binding the L-strand (L1 and L2) and two TALEs binding the H-strand (H1 and H2) of the mtDNA were designed to yield variable spacing windows, so that the target cytosine will be positioned at the center to reflect the preference of the DddA$_{tox}$ G1333 split. **c**. Schematic of the general workflow for the screening of the MitoKO library involving transient transfection of cultured mouse NIH/3T3 cells with plasmids co-expressing DdCBE monomers and fluorescent marker proteins, followed by FACS-based selection of cells expressing both monomers and evaluation of mtDNA from DdCBE-treated cells, 7 days post-transfection.

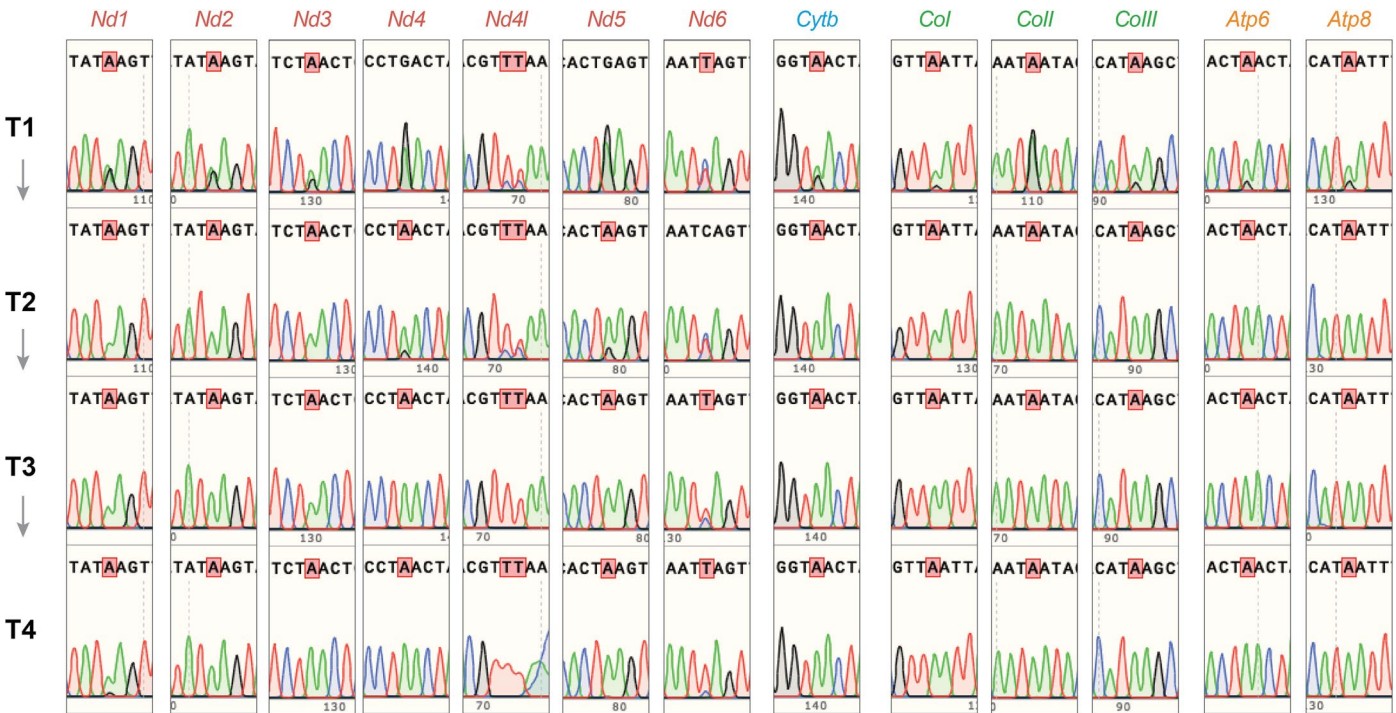

**Extended Data Fig. 2 | Sequential transfection of MitoKO constructs.** Editing of each mouse mtDNA protein-encoded gene (Nd1, Nd2, Nd3, Nd4, Nd4l, Nd5, Nd6, Cytb, CoI, CoII, CoIII, Atp6 and Atp8) using the MitoKO library in the indicated time-points as in Fig. 4 (T1, T2, T3 and T4), analyzed by Sanger sequencing.

**a**

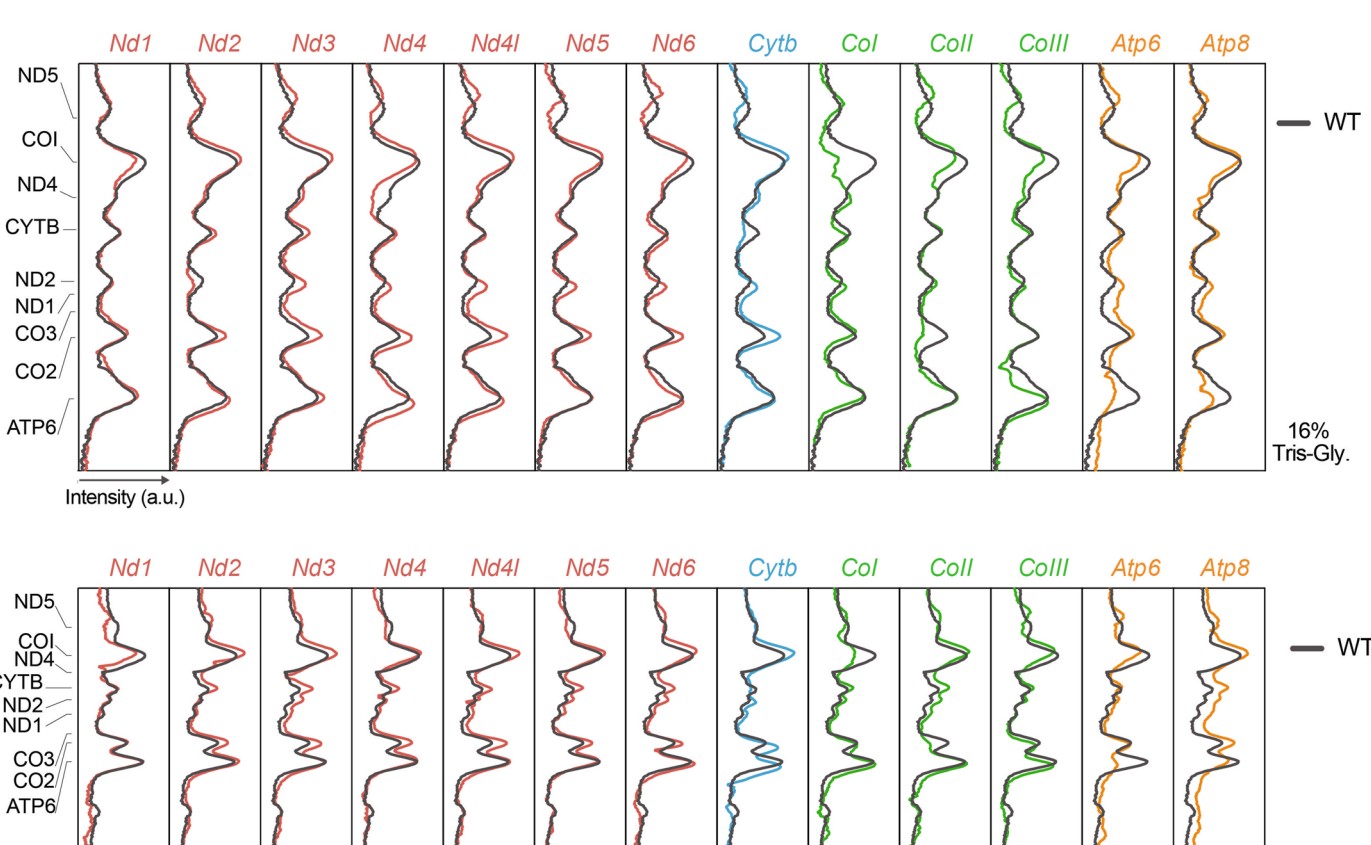

**b**

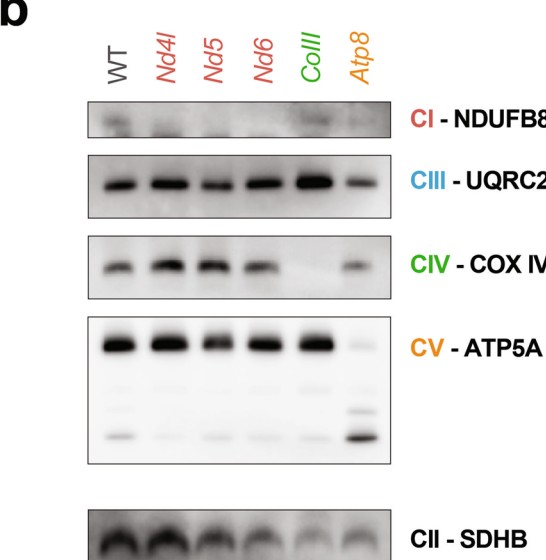

**Extended Data Fig. 3 | See next page for caption.**

**Extended Data Fig. 3 | Mitochondrial translation and OXPHOS complex integrity in the MitoKO DdCBE-treated cells. a**. Densitometric quantification of the mitochondrial de novo protein synthesis gels presented in Fig. 4c. WT (dark grey line) was used as baseline control in each mtDNA protein-encoded gene. The top panel refers to the 16% Tris-Gly gel and the bottom panel refers to the 16% Tricine gel. Source data are provided as a Source Data file. **b**. Immunoblotting of a BNGE with antibodies specific to each mitochondrial complex: NDUFB8 (CI), UQRC2 (CIII) COX IV (CIV) and ATP5A (CV), in mitochondria isolated from NIH/3T3 cells that underwent four rounds of MitoKO DdCBE treatment with the Nd4l, Nd5, Nd6, CoIII, Atp8 pairs and WT controls. An antibody specific for complex II (SDHB) was used as loading control. Uncropped scans are provided as a Source Data file.

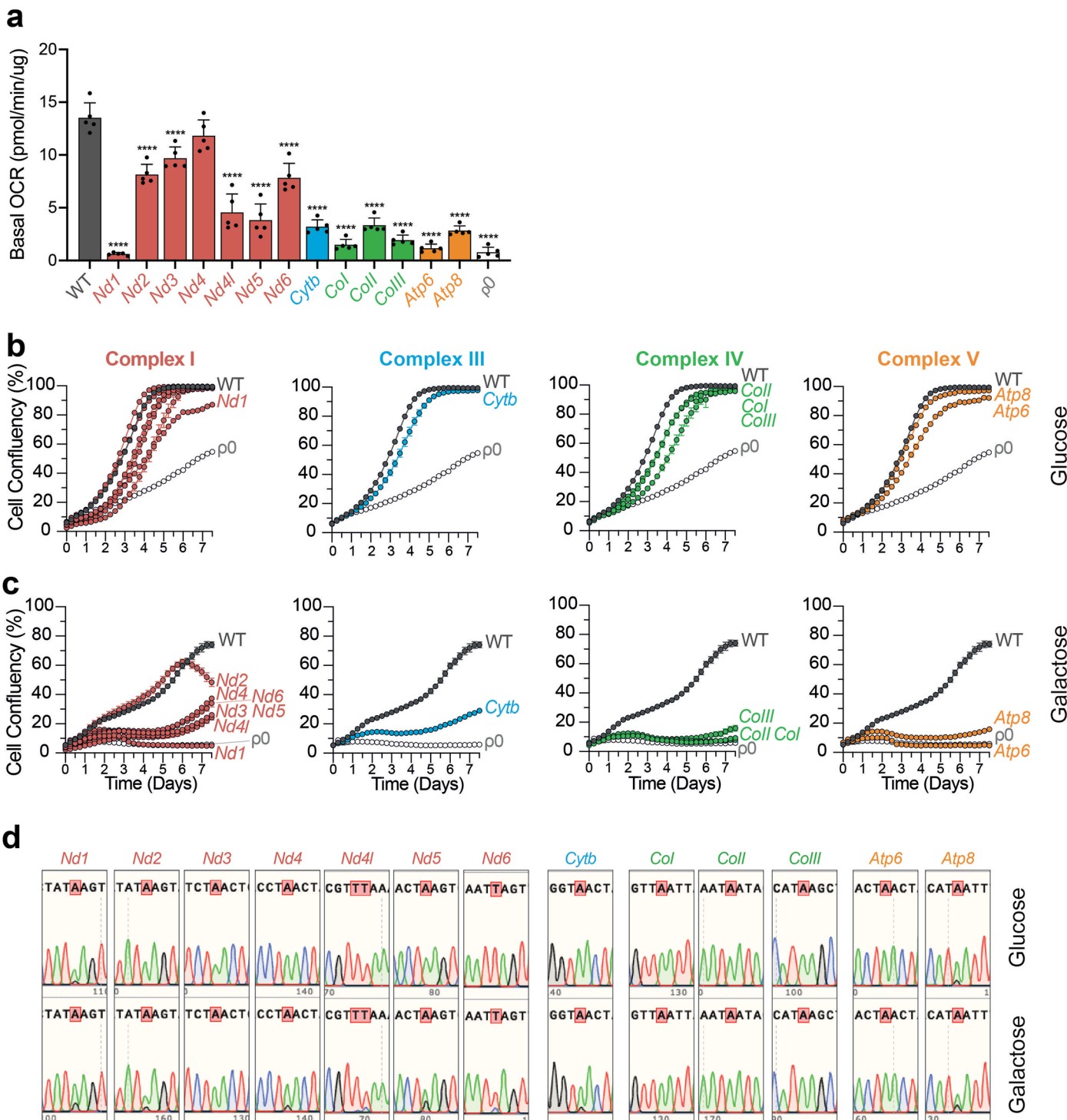

**Extended Data Fig. 4 | Mitochondrial respiration and cell growth of MitoKO DdCBE-treated cells. a**. Basal oxygen consumption rates (OCRs) of MitoKO DdCBE-treated cells after 4 cycles of iterative treatments (see Fig. 4). Wild-type (WT) and p0 cells, containing no mtDNA, were used as controls. Bars represent the mean and error bars represent ± SEM (n = 5, biological replicates). Ordinary one-way ANOVA with Dunnett´s test: **** P-value < 0.0001. Source data are provided as a Source Data file. **b,c**. Growth curves of MitoKO DdCBE-treated cells after 4 cycles of iterative treatments (see Fig. 4) cultured in DMEM supplemented with either glucose (a) or galactose (b). Galactose necessitates mitochondrial ATP

production via oxidative phosphorylation. Each cell line is indicated on the plot and grouped by mitochondrial respiratory complex. Wild-type (WT) and p0 cells, containing no mtDNA, were used as controls. Measurements were done using an Incucyte S3 live-cell imaging system. 9 images were taken per well every 6 hours for 7.5 days. Coloured dots represent mean (n = 3, biological replicates). Source data are provided as a Source Data file. **d**. Editing of MitoKO DdCBE-treated cells after being cultured for 7.5 days in DMEM supplemented with either glucose or galactose, analyzed by Sanger sequencing.

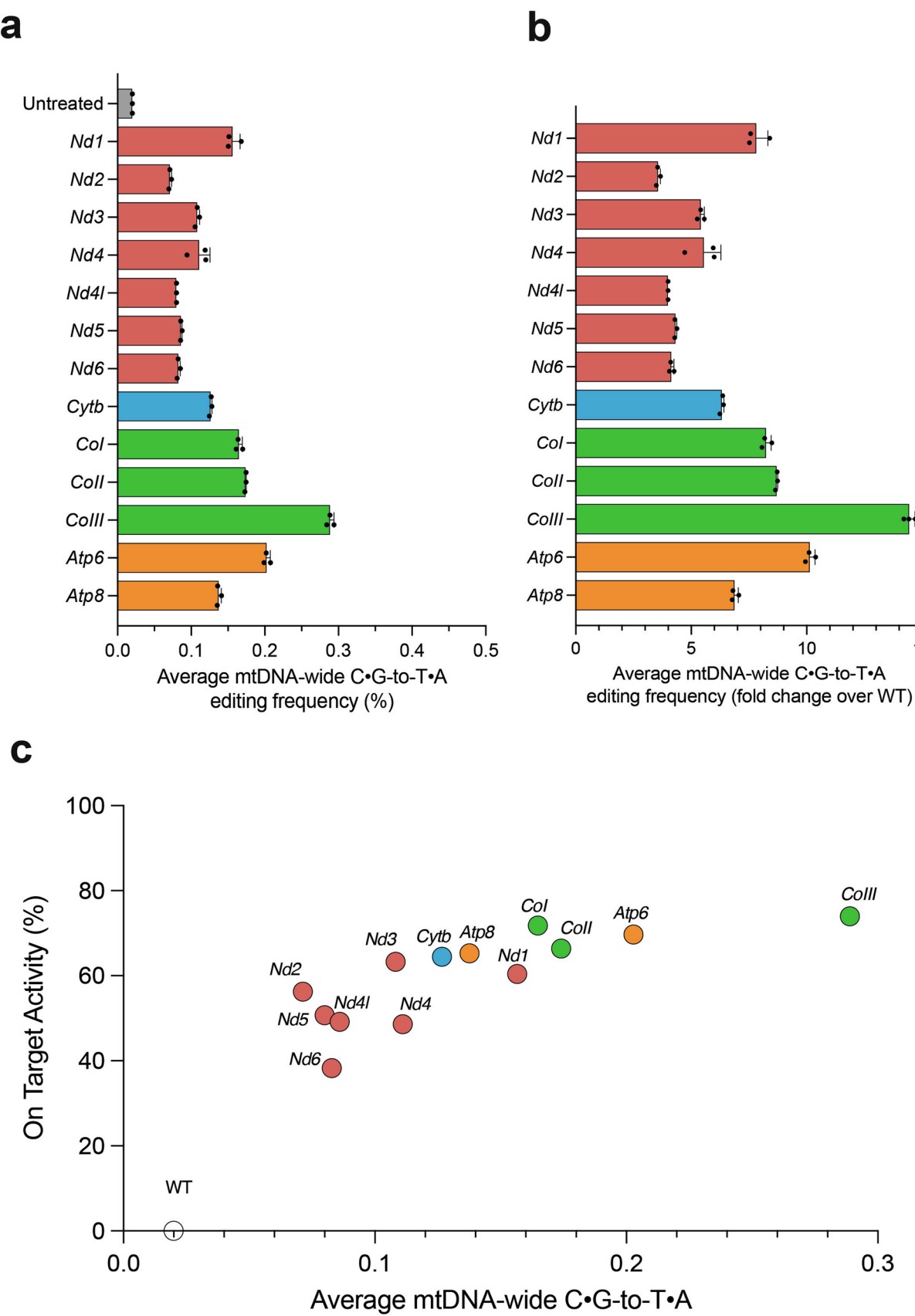

**Extended Data Fig. 5 | See next page for caption.**

**Extended Data Fig. 5 | mtDNA off-target editing by the lead MitoKO DdCBE pairs following high level expression. a,b**. Mitochondrial genome-wide off-targeting of the lead MitoKO DdCBE pairs (Fig. 3) measured by NGS 14 days after transfection and presented as absolute values (a) or fold change over wild-type (WT) controls (b). Bars represent the mean and error bars represent ± SEM (n = 3, technical replicates). Source data are provided as a Source Data file. **c**. On-target (Y axis) vs. off-target (X axis) for the lead MitoKO DdCBE pairs. Dots represent the mean (n = 3, technical replicates). Source data are provided as a Source Data file.

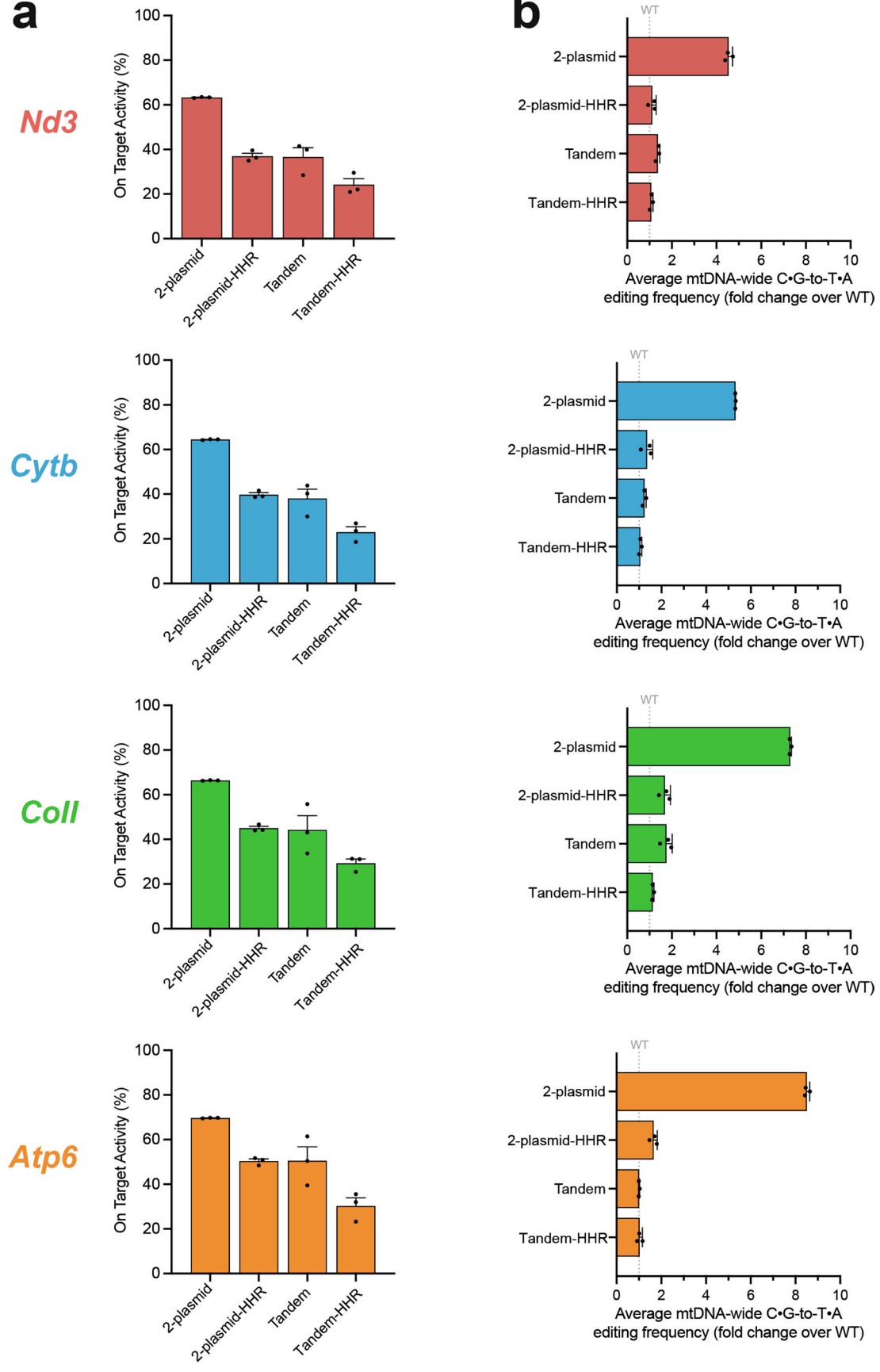

**Extended Data Fig. 6 | See next page for caption.**

**Extended Data Fig. 6 | mtDNA on and off-target editing by the lead MitoKO DdCBE pairs following fine-tuned expression. a**. On-target performance of the *Nd3*, *Cytb*, *CoII* and *Atp6* MitoKO constructs transiently delivered into NIH/3T3 cells, as separate monomers (2-plasmid), separate monomers with the hammerhead ribozyme (HHR) incorporated in mRNA (2-plasmid-HHR), bi-cistronic construct, with the tandemly arrayed DdCBE monomers linked by the T2A element (Tandem) and the tandem, T2A-linked monomers harboring HHR (Tandem-HHR). Bars represent the mean and error bars represent ± SEM (n = 3, biological replicates). Source data are provided as a Source Data file. **b**. Mitochondrial genome-wide off-targeting the *Nd3*, *Cytb*, *CoII* and *Atp6* MitoKO constructs measured by NGS and presented fold change over wild-type (WT) controls. Bars represent the mean and error bars represent ± SEM (n = 3, biological replicates). Source data are provided as a Source Data file.

**a**

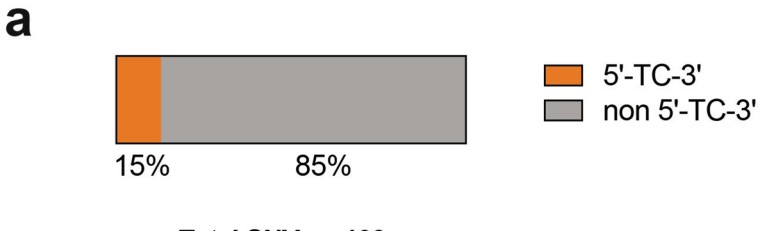

**b**

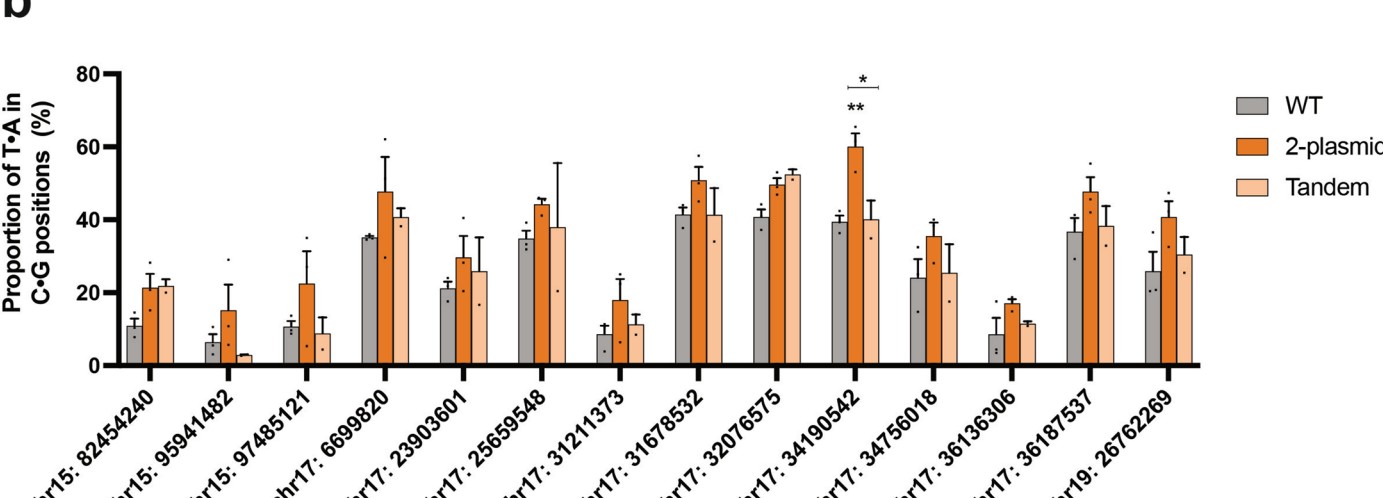

**c**

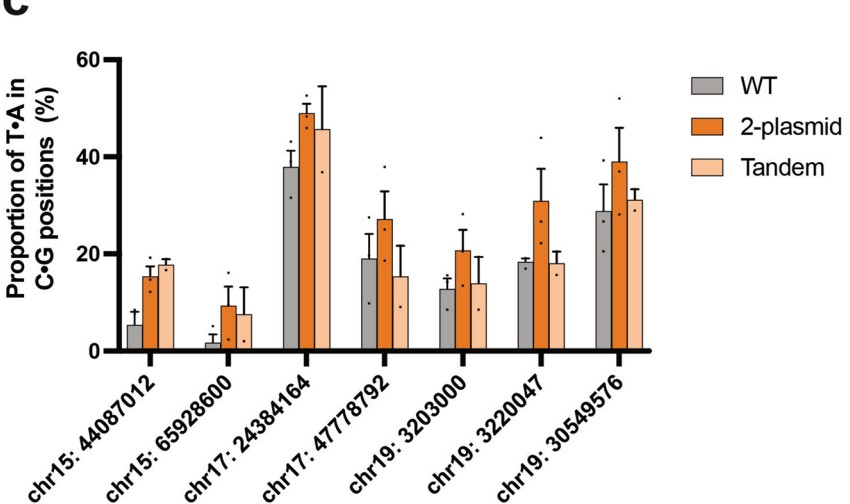

**Extended Data Fig. 7 | Nuclear DNA off-target analysis by the Atp6 MitoKO DdCBE pair. a**. Number of SNVs identified in chromosomes 15 to 19, where T:A nucleotides are identified in C:G positions in cells transfected with the 2-plasmid version of the *Atp6* MitoKO constructs, as compared to WT cells, and the proportion of SNVs found in a 5´-TC-3´ context or non 5´-TC-3´ context. **b,c**. NGS analysis of the proportion of T:A identified in C:G loci at single positions of chromosomes 15 to 19 that are in a (b) 5´-TC-3´ or (c) 5´-TC<u>C</u>-3´ context. Cells were transiently transfected (7 days) with the 2-plasmid or tandem architecture of the *Atp6* MitoKO constructs and compared with WT controls. Bars represent the mean and error bars represent ± SEM (n = 3 for WT and 2-plasmid, n = 2 for Tandem, biological replicates). Ordinary two-way ANOVA with Tukeys´s test: * P-value < 0.05; ** P-value < 0.01. Source data are provided as a Source Data file.

# Reporting Summary

## Statistics

For all statistical analyses, confirm that the following items are present in the figure legend, table legend, main text, or Methods section.

| n/a | Confirmed | |
|---|---|---|
| ☐ | ☒ | The exact sample size (*n*) for each experimental group/condition, given as a discrete number and unit of measurement |
| ☐ | ☒ | A statement on whether measurements were taken from distinct samples or whether the same sample was measured repeatedly |
| ☐ | ☒ | The statistical test(s) used AND whether they are one- or two-sided<br>*Only common tests should be described solely by name; describe more complex techniques in the Methods section.* |
| ☒ | ☐ | A description of all covariates tested |
| ☒ | ☐ | A description of any assumptions or corrections, such as tests of normality and adjustment for multiple comparisons |
| ☐ | ☒ | A full description of the statistical parameters including central tendency (e.g. means) or other basic estimates (e.g. regression coefficient) AND variation (e.g. standard deviation) or associated estimates of uncertainty (e.g. confidence intervals) |
| ☒ | ☐ | For null hypothesis testing, the test statistic (e.g. *F*, *t*, *r*) with confidence intervals, effect sizes, degrees of freedom and *P* value noted<br>*Give P values as exact values whenever suitable.* |
| ☒ | ☐ | For Bayesian analysis, information on the choice of priors and Markov chain Monte Carlo settings |
| ☒ | ☐ | For hierarchical and complex designs, identification of the appropriate level for tests and full reporting of outcomes |
| ☒ | ☐ | Estimates of effect sizes (e.g. Cohen's *d*, Pearson's *r*), indicating how they were calculated |

*Our web collection on statistics for biologists contains articles on many of the points above.*

## Software and code

Policy information about availability of computer code

| Data collection | Illumina MiSeq manufacturer's software (version 4.0) |
|---|---|
| Data analysis | TrimGalore! version 0.6.6 (doi: 10.5281/zenodo.5127899)<br>https://github.com/FelixKrueger/TrimGalore/releases<br>Bowtie version 2 (doi: 10.1038%2Fnmeth.1923)<br>https://github.com/BenLangmead/bowtie2<br>Samtools version 1.12 (doi: 10.1093/gigascience/giab008)<br>https://github.com/samtools/samtools<br>HiSat2 version 2.2.1<br>http://daehwankimlab.github.io/hisat2/<br>VarScan version 2 (doi: 10.1101/gr.129684.111)<br>https://github.com/Jeltje/varscan2<br>Cutadapt version 3.5 (doi: 10.14806/ej)<br>https://github.com/marcelm/cutadapt<br>REDItools 2.0 (doi.org/10.1186/s12859-020-03562-x)<br>https://github.com/tizianoflati/reditools2.0<br>Graphpad Prism 9 for macOS version 9.3.1<br>Microsoft Excel version 15.32 |

For manuscripts utilizing custom algorithms or software that are central to the research but not yet described in published literature, software must be made available to editors and reviewers. We strongly encourage code deposition in a community repository (e.g. GitHub). See the Nature Portfolio guidelines for submitting code & software for further information.

## Data

Policy information about availability of data

All manuscripts must include a data availability statement. This statement should provide the following information, where applicable:
- Accession codes, unique identifiers, or web links for publicly available datasets
- A description of any restrictions on data availability
- For clinical datasets or third party data, please ensure that the statement adheres to our policy

> The data supporting the findings of this study are available within the paper and its Supplementary Information. Source data for the figures are provided with this paper. The NGS files generated in this study are available from the GEO database via the accession number GSE202643.

# Field-specific reporting

Please select the one below that is the best fit for your research. If you are not sure, read the appropriate sections before making your selection.

☒ Life sciences ☐ Behavioural & social sciences ☐ Ecological, evolutionary & environmental sciences

For a reference copy of the document with all sections, see nature.com/documents/nr-reporting-summary-flat.pdf

# Life sciences study design

All studies must disclose on these points even when the disclosure is negative.

| | |
|---|---|
| Sample size | No statistical methods were used to predetermine sample sizes for the in vitro and in vivo experiments. Sample sizes were chosen on the basis of existing procedures and standards in the field and on the basis of biological replicates (with n values of at least 2). Whenever experimentally possible and reasonable, more replicates were performed. |
| Data exclusions | No data were excluded from the analyses. |
| Replication | Experiments with duplicates or triplicates of cells were done with distinct aliquots of cells at intervals of at least one week. A biological replicate in the in vivo experiments corresponds to an individual mouse. All experiments were repeated at least once. All attempts at replication were successful. |
| Randomization | To each DdCBE pair an ID number was attributed in a numerical order. Cells were treated with DdCBE pairs also in a numerical order. |
| Blinding | Samples were collected and analysed by NGS by independent researchers, without any description. |

# Reporting for specific materials, systems and methods

We require information from authors about some types of materials, experimental systems and methods used in many studies. Here, indicate whether each material, system or method listed is relevant to your study. If you are not sure if a list item applies to your research, read the appropriate section before selecting a response.

## Materials & experimental systems

| n/a | Involved in the study |
|---|---|
| ☐ | ☒ Antibodies |
| ☐ | ☒ Eukaryotic cell lines |
| ☒ | ☐ Palaeontology and archaeology |
| ☐ | ☒ Animals and other organisms |
| ☒ | ☐ Human research participants |
| ☒ | ☐ Clinical data |
| ☒ | ☐ Dual use research of concern |

## Methods

| n/a | Involved in the study |
|---|---|
| ☒ | ☐ ChIP-seq |
| ☒ | ☐ Flow cytometry |
| ☒ | ☐ MRI-based neuroimaging |

## Antibodies

| | |
|---|---|
| Antibodies used | Primary antibodies used:<br>1. mouse anti-NDUFB8, dilution 1:1000 (CN Abcam, ab110242)<br>2. mouse anti-SDHB, dilution 1:2000 (CN Abcam, ab14714)<br>3. mouse anti-UQRC2, dilution 1:1000 (CN Abcam, ab14745)<br>4. mouse anti-COX IV, dilution 1:1000 (CN Abcam, ab14744)<br>5. mouse anti-ATP5A, dilution 1:1000 (CN Abcam, ab14748) |

| Validation | Secondary antibodies used: |
| --- | --- |
| | 6. HRP-linked Goat Anti-Mouse IgG (Promega, W4021) - Western blot |
| | Validation statements in the manufacturer's website: |
| | 1. mouse anti-NDUFB8: "Validated in WB, IHC and tested in Mouse, Rat, Cow, Human". |
| | 2. mouse anti-SDHB: "Validated in WB, IHC, Flow Cyt, ICC/IF and tested in Mouse, Rat, Cow, Human". |
| | 3. mouse anti-UQRC2: "Validated in WB, IHC, Flow Cyt and tested in Human". Validated in mouse in this study. |
| | 4. mouse anti-COX IV: "Validated in WB, Flow Cyt and tested in Mouse, Rat, Cow, Human". |
| | 5. mouse anti-ATP5A: "Validated in WB, IHC-P, ICC/IF, Flow Cyt and tested in Mouse, Rat, Cow, Human, Drosophila melanogaster" |

## Eukaryotic cell lines

Policy information about cell lines

| Cell line source(s) | NIH/3T3 (ATCC; CRL-1658); Flp-In-3T3 (ThermoFisher, cn. R76107); 3T3 rho-0#8 (Kerafast, cn. ESA101) |
| --- | --- |
| Authentication | Cell lines were not authenticated for this study. |
| Mycoplasma contamination | The cell medium was tested for mycoplasma contamination. All tests were negative. |
| Commonly misidentified lines (See ICLAC register) | No commonly misidentified cell lines were used. |

## Animals and other organisms

Policy information about studies involving animals; ARRIVE guidelines recommended for reporting animal research

| Laboratory animals | Mice (Mus Musculus) in a C57BL/6J background were obtained from Charles River Laboratories. The animals were maintained in a temperature-and humidity-controlled animal-care facility with a 12-h-light/12-h-dark cycle and free access to water and food, and they were sacrificed by cervical dislocation. |
| --- | --- |
| Wild animals | The study did not involve wild animals. |
| Field-collected samples | The study did not involve samples collected from the field. |
| Ethics oversight | All animal experiments were carried out in accordance with the UK Animals (Scientific Procedures) Act 1986 (PPL: P6C20975A) and the EU Directive 2010/63/EU. |

Note that full information on the approval of the study protocol must also be provided in the manuscript.

