## [Peer Review File · Nature Biomedical Engineering]

A library of base editors for the precise ablation of all protein-coding genes in the mouse mitochondrial genome

Corresponding author: Michal Minczuk

Editorial note

This document includes relevant written communications between the manuscript's corresponding author and the editor and reviewers of the manuscript during peer review. It includes decision letters relaying any editorial points and peer-review reports, and the authors' replies to these (under 'Rebuttal' headings). The editorial decisions are signed by the manuscript's handling editor, yet the editorial team and ultimately the journal's Chief Editor share responsibility for all decisions.

Any relevant documents attached to the decision letters are referred to as **Appendix #**, and can be found appended to this document. Any information deemed confidential has been redacted or removed. Earlier versions of the manuscript are not published, yet the originally submitted version may be available as a preprint. Because of editorial edits and changes during peer review, the published title of the paper and the title mentioned in below correspondence may differ.

Correspondence

Sat 02 Jul 2022

Decision on Article nBME-22-1287

Dear Dr Minczuk,

Thank you again for submitting to *Nature Biomedical Engineering* your manuscript, "MitoKO: A library of base editors for precise ablation of all protein-coding genes in the mouse mitochondrial genome". The manuscript has been seen by three experts, whose reports you will find at the end of this message. You will see that the reviewers appreciate the work, and that Reviewers #1 and #3 raise a relatively small number of technical criticisms that I hope you will be able to address.

When you are ready to resubmit your manuscript, please upload the revised files, a point-by-point rebuttal to the comments from all reviewers, the reporting summary, and a cover letter that explains the main improvements included in the revision and responds to any points highlighted in this decision.

Please follow the following recommendations:

- * Clearly highlight any amendments to the text and figures to help the reviewers and editors find and understand the changes (yet keep in mind that excessive marking can hinder readability).
- * If you and your co-authors disagree with a criticism, provide the arguments to the reviewer (optionally, indicate the relevant points in the cover letter).
- * If a criticism or suggestion is not addressed, please indicate so in the rebuttal to the reviewer comments and explain the reason(s).
- * Consider including responses to any criticisms raised by more than one reviewer at the beginning of the rebuttal, in a section addressed to all reviewers.* The rebuttal should include the reviewer comments in point-by-point format (please note that we provide all reviewers will the reports as they appear at the end of this message).

* Provide the rebuttal to the reviewer comments and the cover letter as separate files.

We hope that you will be able to resubmit the manuscript within 12 weeks from the receipt of this message. If this is the case, you will be protected against potential scooping. Otherwise, we will be happy to consider a revised manuscript as long as the significance of the work is not compromised by work published elsewhere or accepted for publication at *Nature Biomedical Engineering*.

We hope that you will find the referee reports helpful when revising the work, which we look forward to receive. Please do not hesitate to contact me should you have any questions.

Best wishes,

Pep

Pep Pàmies
Chief Editor, Nature Biomedical Engineering

Reviewer #1 (Report for the authors (Required)):

In the manuscript, the authors developed a toolset to knock out 13 mtDNA protein coding genes in the mouse mitochondrial genome. Using this toolset, they generated near-complete knock-out cells in vitro. However, in vivo, the editing efficiency in founder mice is much lower than that in cells. It's reported that a premature stop codon was incorporated into mtDNA to generate a mouse model by DdCBE (PMID: 33608520), to some extent, compromising the novelty of this manuscript. Given that this toolset may be useful for mitochondrial disease research, the authors should address the following concerns before publication.

Major

1. In Figure 3C, the protein level of ND4L, ND5, ND6, CO3 and ATP8 did not decrease as markedly as the authors claimed. Western blot should be performed to confirm these results for each protein.
2. The authors should perform TEM and Seahorse assays to provide direct evidence that knocking out each mtDNA protein coding gene affects mitochondrial ultrastructure and function in cells.
3. Although the authors could obtain F3 pups with high heteroplasmy levels by selective breeding, it compromises the feasibility of the toolset to generate mitochondrial disease models. The authors should provide solutions and evidence to generate mice with mutation loads high enough to display phenotypes at multiple sites. Considering too high mutation loads would lead to embryonic lethality, conditional knockout of mtDNA protein coding genes may be a good strategy.
4. In mouse models, potential nuclear off-targets can be eliminated by back-crossing with wild type males. While, in cell lines, how to demonstrate that the phenotype is resulted from on-target editing, rather than nuclear off-target editing. Does DdCBE pair linked by the T2A element decrease nuclear off-targets, given that it could improve the specificity in mitochondria?
5. Except Nd3, Cytb, Co2 and Atp6, do the optimized DdCBE pairs improve the specificity for others mtDNA protein coding genes?

Minor

1. Does a premature stop codon affect the stability of polycistronic RNA in mitochondria?

2. Average mtDNA-wide CCG-to-TTA editing frequency may mask the bona fide off-targets. The editing efficiencies of the top off-target sites detected by the original DdCBE should be shown for the optimized ones.

3. Line 152 and 154, Extended Data Figure 2 should be Extended Data Figure 3.

Reviewer #2 (Report for the authors (Required)):

In this manuscript, Silva-Pinheiro et al report on the development of genetic tools to manipulate mitochondrial DNA in murine cells and the mouse. Expression of mitochondrial DNA contributes a handful of proteins to the oxidative phosphorylation machinery, which is the main machinery for ATP production. So far, directed engineering of the mitochondrial DNA has not been possible, but the usage of bacterial base deaminases is revolutionizing the field of mitochondrial biology and genetics.

In this work, the authors have now established a system to generate mtDNA molecules harboring premature stop codons in distinct ORFs, thereby creating a panel of constructs that allow to abolish synthesis of full length mtDNA-encoded proteins. This study is overall very well conducted, it is very well written and explains the careful analyses done by the group in detail. As it stands, it presents a significant technological advance for the field, which will allow to generate new cell and mouse models for pre-clinical studies on pathomechanisms and similar aspects, as well as detailed mechanistic investigations into various aspects of mitochondrial gene expression and OXPHOS assembly.

Reviewer #3 (Report for the authors (Required)):

This paper describes a contribution of the vector construction for knocking out each gene of all mitochondrial DNA-encoded proteins in mice. The authors developed the methods originally invented by Mok et al, the use of TALE and dDBA with mitochondrial targeting signals to change a base C to T in the targeted mitochondrial DNAs. The high efficiency and specificity (low off-target) with effective depletion of each targeted protein in cultured cells are impressive. Although the targeted genes are essential to the mouse body, they succeeded to have heteroplasmic mice with a substantially high level of mutated DNA ratio. The availability of effective vectors for all proteins will contribute well to the research community, and be suitable for the scope of this journal. The manuscript was well-organized and easy to follow without typos I found. Some revisions according to the comments below will strengthen the MS.

Major points

No

Five minor points

Add the recent paper about the nuclear off-target effect of ddCBE, <https://www.nature.com/articles/s41586-022-04836-5>, (no further discussion is necessarily required.)

Why not add the papers on relatively high efficient editing of plant mitochondrial and plastid genomes, <https://doi.org/10.1073/pnas.2121177119>, <https://doi.org/10.1038/s41477-021-00943-9>, <https://www.nature.com/articles/s41477-021-00954-6>

Describe the possible reasons why the rate of base editing in this paper was so high compared to the previous results like the paper Mok et al.

A short description of the phenotype of the F3 mouse would be helpful. (No phenotype would not be surprised because of the rate of heteroplasmy).

Quantification of fig 3c might be helpful in addition to the good gel images.

Fri 07 Oct 2022

Decision on Article NBME-22-1287A

Dear Dr Minczuk,

Thank you again for your revised manuscript, "MitoKO: A library of base editors for precise ablation of all protein-coding genes in the mouse mitochondrial genome". Having consulted with Reviewers #1 and #3 (whose comments you will find at the end of this message), I am pleased to write that we shall be happy to publish the manuscript in *Nature Biomedical Engineering*.

We will be performing detailed checks on your manuscript, and in due course will send you a checklist detailing our editorial and formatting requirements. You will need to follow these instructions before you upload the final manuscript files.

Best wishes,

Pep

Pep Pàmies
Chief Editor, Nature Biomedical Engineering

Reviewer #1 (Report for the authors (Required)):

The authors have addressed most of my concerns and I may recommend the manuscript for publication before fixing the following minor concerns.

When performing Seahorse assay, besides the basal OCR, what happens to the OCRs of ATP production, proton leak and maximal respiration in 13 mtDNA knock-out cell lines?

I suggest the authors discuss the possibility that performing whole genome sequencing, even greater than 30x coverage, may not be able to detect DdCBE-induced nuclear off-target editing in large pools of cells, because it will result in loss of signal for random off-target effects due to population averaging.

"F0 to F3" should be "F0 to F4" in the title of Supplementary Dataset 3.

Reviewer #3 (Report for the authors (Required)):

The revised manuscript of Silva-Pinheiro et al., "MitoKO: A library of base editors for precise ablation of all protein-coding genes in the mouse mitochondrial genome." is well written and has no additional problems. All of my and other reviewers' comments were replied well. I don't have any additional comments.

Rebuttal 1

Reviewer #1

In the manuscript, the authors developed a toolset to knock out 13 mtDNA protein coding genes in the mouse mitochondrial genome. Using this toolset, they generated near-complete knock-out cells in vitro. However, in vivo, the editing efficiency in founder mice is much lower than that in cells. It's reported that a premature stop codon was incorporated into mtDNA to generate a mouse model by DdCBE (PMID: 33608520), to some extent, compromising the novelty of this manuscript. Given that this toolset may be useful for mitochondrial disease research, the authors should address the following concerns before publication.

Thank you very much for a positive assessment of our work and for suggesting the revision points below. Regarding the paper reporting mtDNA editing in mice you mentioned (which we reference in the introduction), we note that none of the mouse models reported by Lee et al 2020 (PMID: 33608520) had mtDNA heteroplasmy levels which would be considered pathogenic nor contained any off-targeting analysis.

Major

1. In Figure 3C, the protein level of ND4L, ND5, ND6, CO3 and ATP8 did not decrease as markedly as the authors claimed. Western blot should be performed to confirm these results for each protein.

We agree with this remark. However, we note that currently it is not possible to perform this experiment due to no specific antibodies being available for ND4L, ND5, ND6, CO3 or ATP8. The field has been struggling with obtaining such antibodies for decades, but owing to very high hydrophobicity of the mtDNA-encoded proteins this task has not been successful yet. However, to address this point, we analysed the integrity of the assembled complex I (for ND4L, ND5, ND6), complex IV (for CO3) and complex V/ATP synthase (for ATP8) using native gel electrophoresis. We detected substantially reduced assembly and/or stability of these complexes, consistent with the ablation of these essential, mtDNA-encoded subunits and with the previously published data. These data are included in the revised version of the manuscript (page 7/8, Extended Data Figure 3b).

2. The authors should perform TEM and Seahorse assays to provide direct evidence that knocking out each mtDNA protein coding gene affects mitochondrial ultrastructure and function in cells.

We note that the scope of our paper is to provide a toolset enabling researchers (for the first time) to develop models of mtDNA dysfunction. We already provided evidence of mitochondrial dysfunction in the KO cells by showing compromised growth on galactose, which forces them to rely on oxidative phosphorylation for energy production. However, to satisfy the Reviewer's request, we have also measured oxygen consumption rates (OCRs) using a Seahorse extracellular flux analyser, which were reduced in the KO cell lines (page 8, Extended Data Figure 4a). Nonetheless, we feel that further in-depth characterization of the MitoKO cell lines e.g. transmission electron microscopy or mitochondrial ultrastructure analysis, while very interesting, is well beyond the scope of the submission.

3. Although the authors could obtain F3 pups with high heteroplasmy levels by selective breeding, it compromises the feasibility of the toolset to generate mitochondrial disease models. The authors should provide solutions and evidence to

generate mice with mutation loads high enough to display phenotypes at multiple sites. Considering too high mutation loads would lead to embryonic lethality, conditional knockout of mtDNA protein coding genes may be a good strategy.

While we agree that conditional mtDNA heteroplasmy mouse model would be of tremendous value for the field, the generation of such animals would require at least 2 years. As already mentioned in the first paragraph of the discussion section, we plan to produce such models (“In the next stage, we plan to implement the potential of somatic tissue-specific mutagenesis with AAV and Rosa26/TIGRE locus transgenesis for spatiotemporal expression control.”). However, due to time constraints we are reluctant to embark on this task as a part of the current submission.

Having said this, we’d like to point out the full-body mtDNA animal models are also expected be of great value for the field through enabling, for example, studies of tissue distribution of heteroplasmy or mtDNA bottleneck segregation. We note again that the main aim of our paper is not to characterize the mouse, but to provide a standardised set of tools to generate these models by us and others. Nonetheless, to address Reviewer’s comment on obtaining animals “with mutation loads high enough to display phenotypes”, in the revised version of the manuscript, we show that the F4 generation of the MT-ATP6 KO mouse, carrying 65-72% mutant heteroplasmy, has a perturbed stability and/or assembly of ATP synthase, a molecular phenotype also observed in patients harbouring truncating mutations in *MT-ATP6* and suffering from mitochondrial diseases (page 13, Fig. 5e-f).

4. In mouse models, potential nuclear off-targets can be eliminated by back-crossing with wild type males. While, in cell lines, how to demonstrate that the phenotype is resulted from on-target editing, rather than nuclear off-target editing. Does DdCBE pair linked by the T2A element decrease nuclear off-targets, given that it could improve the specificity in mitochondria?

We only partially agree with this remark. Nuclear off-targets in cultured cells can also be avoided by producing cytoplasmic hybrids (cybrids), whereby mitochondria harbouring edited, mutant mtDNA are transferred to mtDNA-less recipient cells that have not undergone DdCBE treatment. Cybridisation of cultured cells is a standard procedure in the mitochondrial field dating back to the eighties (King and Attardi, 1989, PMID: 2814477) with well standardised and accessible protocols (e.g. Cavaliere et al, 2022, PMID: 35343952). We discuss this issue in the revised manuscript (page 18). Nonetheless, to address Reviewer’s comment, we have performed nuclear DNA off-target analysis for the MitoKO Atp6 constructs transfected to cultured cells using a genome-wide NGS approach for control, high-level (2-plasmids) and low level (Tandem) expression conditions. We did not detect significant nDNA off-targeting (Extended Data Figure 7). We discuss these data in the context of recently published papers claiming substantial nDNA off-targeting caused by DdCBEs (page 12).

5. Except Nd3, Cytb, Co2 and Atp6, do the optimized DdCBE pairs improve the specificity for others mtDNA protein coding genes?

Our interpretation of this comment is a request to perform the analysis as presented in Fig. 4 for the remaining nine mtDNA-encoded genes, in addition to the 4 genes already used as proof-of-concept. Of course, it can be done, however, in our view it would be of limited scientific value, as we do not see any reason why the approach that worked for 4 genes would not work for other genes encoded on the same molecule, especially given that all 13 constructs display very comparable properties

(Fig. 2). In addition, this analysis would take extra 6-9 months, substantially delaying the publication of our report.

Minor

1. Does a premature stop codon affect the stability of polycistronic RNA in mitochondria?

This is the type of question others in the field, interested in mitochondrial RNA biology will be able to answer upon publication of the MitoKO library.

2. Average mtDNA-wide C·G-to-T·A editing frequency may mask the bona fide off-targets. The editing efficiencies of the top off-target sites detected by the original DdCBE should be shown for the optimized ones.

We agree with this comment and provide the requested data as Supp. Dataset 2.

3. Line 152 and 154, Extended Data Figure 2 should be Extended Data Figure 3.

Thank you for pointing this out. We have corrected this mistake.

Reviewer #2

In this manuscript, Silva-Pinheiro et al report on the development of genetic tools to manipulate mitochondrial DNA in murine cells and the mouse. Expression of mitochondrial DNA contributes a handful of proteins to the oxidative phosphorylation machinery, which is the main machinery for ATP production. So far, directed engineering of the mitochondrial DNA has not been possible, but the usage of bacterial base deaminases is revolutionizing the field of mitochondrial biology and genetics.

In this work, the authors have now established a system to generate mtDNA molecules harboring premature stop codons in distinct ORFs, thereby creating a panel of constructs that allow to abolish synthesis of full length mtDNA-encoded proteins. This study is overall very well conducted, it is very well written and explains the careful analyses done by the group in detail. As it stands, it presents a significant technological advance for the field, which will allow to generate new cell and mouse models for pre-clinical studies on pathomechanisms and similar aspects, as well as detailed mechanistic investigations into various aspects of mitochondrial gene expression and OXPHOS assembly.

We are grateful for this very positive assessment of our work. We have not identified any addressable criticism by Reviewer 2.

Reviewer #3

This paper describes a contribution of the vector construction for knocking out each gene of all mitochondrial DNA-encoded proteins in mice. The authors developed the methods originally invented by Mok et al, the use of TALE and dDBA with mitochondrial targeting signals to change a base c to T in the targeted mitochondrial DNAs. The high efficiency and specificity (low off-target) with effective depletion of each targeted protein in cultured cells are impressive. Although the targeted genes are essential to the mouse body, they succeeded to have heteroplasmic mice with a substantially high level of mutated DNA ratio. The availability of effective vectors for all proteins will contribute well to the research community, and be suitable for the scope of this journal. The manuscript was well-organized and easy to follow without typos I found. Some revisions according to the comments below will strengthen the MS.

Major points

No

We would like to thank for this very positive assessment of our work.

Five minor points

Add the recent paper about the nuclear off-target effect of ddCBE, <https://www.nature.com/articles/s41586-022-04836-5>, (no further discussion is necessarily required.)

We are now referencing this paper in the context of nDNA off-target analysis requested by Reviewer 1 (page 12).

Why not add the papers on relatively high efficient editing of plant mitochondrial and plastid genomes, <https://doi.org/10.1073/pnas.2121177119>, <https://doi.org/10.1038/s41477-021-00943-9>, <https://www.nature.com/articles/s41477-021-00954-6>

We have added a sentence about plastid genome editing and cited the above references in the introduction.

Describe the possible reasons why the rate of base editing in this paper was so high compared to the previous results like the paper Mok et al.

We have added a sentence to address this issue in the result section (page 7/8)

A short description of the phenotype of the F3 mouse would be helpful. (No phenotype would not be surprised because of the rate of heteroplasmy).

To address Reviewer's comment, we bred the MT-ATP6 KO animals to F4 and show that animals carrying 65-72% mutant heteroplasmy have a perturbed stability and/or assembly of ATP synthase, a molecular phenotype also observed in patients harbouring truncating mutations in *MT-ATP6* and suffering from mitochondrial diseases (page 13, Fig. 5e-f).

Quantification of fig 3c might be helpful in addition to the good gel images.

We are providing a densitometric quantification of the data presented in Fig. 3c as Extended Data Figure 3a.

Other changes to the manuscript.

We are providing protein sequences of all DdCBE construct used in the study.

Rebuttal 2

Below we are addressing the last two minor comments of Reviewer #1.

The authors have addressed most of my concerns and I may recommend the manuscript for publication before fixing the following minor concerns.

We are grateful for recommending our study for publication and are addressing the remaining two minor points below.

When performing Seahorse assay, besides the basal OCR, what happens to the OCRs of ATP production, proton leak and maximal respiration in 13 mtDNA knock-out cell lines?

These parameters MitoKO cell line bioenergetics have not been measured. We note that the scope of our paper is to provide a toolset enabling researchers to develop models of mtDNA dysfunction. We already provided strong evidence for mitochondrial dysfunction in the KO cells by showing compromised mitochondrial translation, lower steady-state levels of OXPHOS complexes, reduced basal OCRs and slow growth rate on galactose as carbon source. We feel that further in-depth characterisation of the MitoKO cell lines as suggested by the Reviewer, while very interesting, remains beyond the scope of the submission.

I suggest the authors discuss the possibility that performing whole genome sequencing, even greater than 30x coverage, may not be able to detect DdCBE-induced nuclear off-target editing in large pools of cells, because it will result in loss of signal for random off-target effects due to population averaging.

Thank you for this suggestion. We have expanded the discussion to address this point as follows: "While we did not observe any substantial nDNA off-targets for MitoKO Atp6 constructs (Extended Data Fig. 7), these previous results highlight the necessity of extensive DdCBE optimization and development of methods allowing for detection of DdCBE-induced nuclear off-target editing in large cell populations."

"F0 to F3" should be "F0 to F4" in the title of Supplementary Dataset 3.

Thank you. This has been corrected.